# EM-Network: Learning Better Latent Variable for Sequence-to-Sequence Models

## Abstract

In a sequence-to-sequence (seq2seq) framework, the use of an unobserved latent variable, such as latent alignment and representation, is important to address the mismatch problem between the source input and target output sequences. Existing seq2seq literature typically learns the latent space by only consuming the source input, which might produce a sub-optimal latent variable for predicting the target. In this paper, we introduce EM-Network that can yield the promising latent variable by leveraging the target sequence as the model's additional training input. The target input is used as guidance to provide the target-side context and reduce the candidates of the latent variable. The proposed framework is trained in a new self-distillation setup, allowing the original sequence model to benefit from the latent variable of the EM-Network. Specifically, the EM-Network's prediction serves as a soft label for training the inner sequence model, which only takes the source as input. We conduct comprehensive experiments on two types of seq2seq models: connectionist temporal classification (CTC) for speech recognition and attention-based encoder-decoder (AED) for machine translation. Experimental results demonstrate that the EM-Network significantly advances the current state-of-the-art approaches. It improves over the best prior work on speech recognition and establishes state-of-the-art performance on WMT'14 and IWSLT'14 datasets.

## 1 Introduction

Throughout the literature on deep learning, sequence-to-sequence (seq2seq) learning has achieved great success in a wide range of applications, especially in speech and natural language processing. Given a source-target pair $(x,y)$, a task of the seq2seq learning is to learn a function for mapping a source sequence $x$ to a target sequence $y$, which generally suffers from source-target mismatch problems, e.g., unequal length, different domain, and modality mismatch. To deal with this issue, learning the latent variable $z$ and how to improve its quality are deemed critically important in sequence modeling. For example, in automatic speech recognition (ASR), the connectionist temporal classification (CTC) (Graves et al., 2006) model defines the latent alignment $z$ to learn the mapping between the speech feature $x$ and the word sequence $y$, as shown in Figure 1. For the counterpart speech synthesis task, some studies (Kim et al., 2020; 2021) have proposed learning an internal alignment $z$ between the text $x$ and the corresponding speech $y$. In natural language processing (NLP), BERT family models (Devlin et al., 2019; Liu et al., 2019; Lan et al., 2020) employ masked language modeling (MLM) to learn the contextualized representations $z$, where the randomly masked tokens are predicted given the context of other tokens. From the perspective of $z$, the self-supervised learning (SSL)-based pre-training enables the model to obtain the robust latent representation $z$ from the input $x$, which offers the desired performance for predicting the target $y$ on the downstream task.

However, it is difficult to learn the optimal latent variable for the learning task. For example, in the case of ASR, CTC models often converge to sub-optimal alignment distributions and produce over-confident predictions (Liu et al., 2018; Yu et al., 2021; Miao et al., 2015). Since there is an exponential number of possible alignment paths, and the alignment information between source and target sequences is rarely available during training, settling on the optimal alignment is quite challenging. From the feature perspective, powerful representation is important to achieve the desired performance. For the recent NLP studies, including machine translation (MT), a large pre-trained

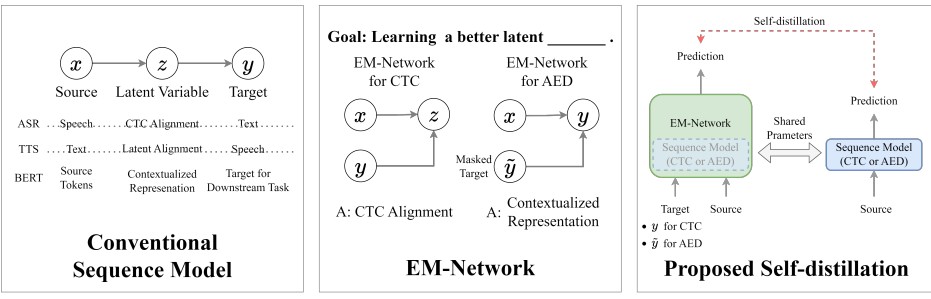

Figure 1: **(Left)** Conventional sequence model converts the source $x$ into the target $y$ through the latent variable $z$. **(Middel)** EM-Network can estimate the promising latent variable $z$ by using the additional training input $y$. For the AED model, the masked version of the target $\tilde{y}$ is used instead of the whole target $y$. **(Right)** EM-Network is trained with a self-distillation setup, where EM-Network's predcition serves as soft labels for training the original sequence model.

LM is highly required to obtain the contextualized representation (Zhu et al., 2020; Wu et al., 2021; Xu et al., 2021). In this paper, we propose a novel framework termed EM-Network that can effectively improve the quality of the latent variable $z$ and thus the overall quality of the seq2seq model. In particular, EM-Network encapsulates two key components. First, the proposed framework leverages the target sequence as the model's additional training input, where the target input $y$ is used as guidance to capture the target-side context and reduce the candidates of $z$. Second, based on the usage of the target input, we present a new self-distillation strategy as collaborative training, where the original sequence model can benefit from the EM-Network's knowledge in a one-stage manner. The prediction of the EM-Network serves as a soft label for training the inner sequence model, which consumes only the source input. The proposed self-distillation acts as a sort of regularization for the seq2seq model, and its performance gain comes from a deep mutual learning scheme (Zhang et al., 2018), where the students learn collaboratively and teach each other. However, the main difference from the previous mutual learning approaches (Zhang et al., 2018; Liang et al., 2021) is that the proposed method utilizes the target input when training the EM-Network instead of merely considering the ground truth as the sole target in training. Since the target input is used as guidance to provide the target-side information, the prediction of the EM-Network (teacher mode) is more accurate than that of the inner sequence model (student mode). Therefore, the sequence model can effectively benefit from the soft labels of the EM-Network, which will be additionally discussed in Section 6. In addition, we attempt to apply the proposed self-distillation to the CTC framework, an unexplored area in mutual learning research.

Modeling the conditional probability of the EM-Network (teacher mode) is determined by whether the latent variable is explicitly defined, as shown in Figure 1. The CTC computation adopts the alignment $z$, and it is difficult to settle on the optimal alignment with the conventional framework. The proposed EM-Network computes the posterior $P(z|x, y)$ for the loss, which aims at predicting a better CTC alignment $z$ by leveraging the source and target inputs. Therefore, the CTC model distilled from the EM-Network does not have to consider the exponential number of possible CTC alignments. We theoretically show that the proposed objective function can serves as the proposed Q-function and is justified from the EM-like algorithm perspective.

For the attention-based encoder-decoder (AED), where there is no explicit latent alignment, it is challenging to directly apply the same training scheme as the EM-Network for CTC. Simply taking the target $y$ as the additional input may cause an obvious but trivial solution, where the model converges with the conditional probability $P(y|x, y) = \delta(y)$. Inspired by the MLM, we present an alternative that employs the masked version of target $\tilde{y}$ as the additional input instead of using the whole target $y$. The EM-Network for AED computes the posterior $P(\tilde{y}|x, y)$ for loss and provides more robust contextualized representations that can benefit the learning task.

We conduct comprehensive experiments on multiple benchmarks, including ASR and MT tasks. The CTC and AED models are considered for ASR and MT tasks, respectively. Experimental results demonstrate that the EM-Network improves over the best prior work on ASR and establishes SOTA performance on WMT'14 and IWSLT'14 datasets.

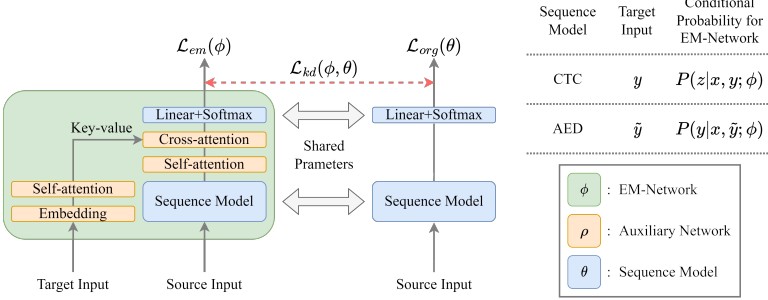

Figure 2: Overview of the EM-Network. Note that the EM-Network includes the parameters of the sequence model. Green components ($\phi$), orange components ($\rho$), blue components ($\theta$) represent the parameters of the EM-Network, the auxiliary network, and the sequence model, respectively. When the sequence model is based on the AED, teacher forcing is applied to the inner sequence model during the training.

## 2 METHODOLOGY

In this section, we introduce how to design the EM-Network for learning better latent variable $z$ (Section 2.1 and Section 2.2), and its applications with CTC (Section 2.3) and AED (Section 2.4). We further provide a theoretical interpretation of the proposed framework from an EM-like algorithm perspective in Section 3.

### 2.1 THE ARCHITECTURE OF EM-NETWORK

**EM-Network.** As shown in Figure 2, EM-Network mainly consists of two parts: (1) a sequence model that performs the original learning task given the source input $x$, and (2) an auxiliary network to learn a meaningful representation of both source $x$ and target $y$. We add the auxiliary network on top of the conventional sequence model. For the convenience of notation, we let $\phi$, $\theta$, and $\rho$ denote the parameters of the EM-Network, the sequence model, and the auxiliary network, respectively. Note that the EM-Network includes the parameters of the sequence model, where $\theta \subset \phi$.

**Sequence Model.** Since the sequence model in the EM-Network performs the same task as the conventional seq2seq models, we can flexibly select the architecture of the sequence model depending on the task demands. For example, in our ASR experiments, the conventional CTC model consisting of 12 transformer (Vaswani et al., 2017) blocks, is applied as the sequence model. For the MT task, the architecture of the sequence model follows that of the BiBERT (Xu et al., 2021).

**Auxiliary Network.** The auxiliary network performs a fusion of the source $x$ and target $y$ representations when generating the prediction, and its architecture is based on the self and cross-attention modules. Specifically, the cross-attention takes the representation of $x$ as the query and the representation of $y$ as the key-value pairs.

### 2.2 TRAINING AND INFERENCE PROCEDURES

**Self-distillation.** EM-Network first yields the predictions, which serve as soft labels in distillation, by encoding both the source $x$ and target $y$ inputs (teacher mode). Then, the sequence model learns the parameter $\theta$ to predict the soft labels given the source input $x$ (student mode).

**Training.** In the proposed framework, there are three kinds of losses during the training phase: (1) the original sequence learning loss $\mathcal{L}_{org}(\theta)$, such as CTC loss and frame-wise cross entropy (CE), with the source input $x$ to train the sequence model, (2) the proposed sequence learning loss $\mathcal{L}_{em}(\phi)$ given both source and target inputs for training the EM-Network, and (3) self-distillation loss $\mathcal{L}_{kd}(\phi, \theta)$ that trains the sequence model to predict the soft labels generated from the EM-Network. We optimize the following loss to train the proposed framework:

$$\mathcal{L}_{train}(\phi, \theta) = \mathcal{L}_{org}(\theta) + \mathcal{L}_{em}(\phi) + \alpha \mathcal{L}_{kd}(\phi, \theta) \tag{1}$$

where $\alpha$ is a tunable parameter. In our self-distillation manner, the soft labels from the EM-Network are updated in each iteration since $\phi$ is affected by the two losses $\mathcal{L}_{em}(\phi)$ and $\mathcal{L}_{kd}(\phi, \theta)$.

**Inference.** During the inference, *only the sequence model is used to generate the prediction*. Since the auxiliary network is removed during the inference, the additional computational load of $\rho$ is only required for the training procedure. In our SSL-based ASR experiments, about 10 M parameters are used as the auxiliary parameters to train the EM-Network, where the number of parameters of $\theta$ is about 93 M. For the fully-supervised ASR setting, the number of parameters of $\theta$ is about 13 M, and only 1 M parameters are required for the auxiliary network.

As aforementioned in Section 1, EM-Network can cover two popular loss functions for sequence learning: CTC and CE objective. We begin in the CTC objective setting (Secion 2.3) and then turn to the CE-based EM-Network (Section 2.4).

## 2.3 EM-NETWORK FOR CTC

**Standard CTC Objective.** Before describing the proposed method in the CTC framework (Graves et al., 2006), it might be beneficial to briefly discuss the CTC objective function. Given an input sequence $x$, the core idea of CTC is leveraging intermediate alignment $z$ by allowing label repetitions possibly interleaved with a special *blank* token ($\epsilon$). CTC trains the sequence model $\theta$ to minimize the following loss function:

$$\mathcal{L}_{org-ctc}(\theta) = -\log P(y|x;\theta) = -\log \sum_{y=\mathcal{B}(z)} P(z|x;\theta) \tag{2}$$

where $\mathcal{B}$ is a many-to-one mapping in the CTC algorithm with $y = \mathcal{B}(z)$. $\mathcal{B}^{-1}(y)$ is the set of possible alignments $z$ compatible to $y$. The mapping is done by merging repeated labels from the paths and then removing all *blank* tokens ($\epsilon$), e.g., $\mathcal{B}(\{\epsilon,a,a,\epsilon,\epsilon,a,b,b\}) = \{aab\}$) where $\{aab\}$ denotes $y$ and the sequence $\{\epsilon,a,a,\epsilon,\epsilon,a,b,b\}$ represents $z$.

**Modeling Conditional Probability with Target Input.** As mentioned earlier, the crucial challenge is how to use the target input while preventing the trivial solution. Since the CTC algorithm explicitly adopts the latent alignment $z$, we can easily apply the target $y$ as the EM-Network's input. The relationship between $z$ and $y$ is many-to-one, which can be formulated as $\mathcal{B}(z) = y$. Intuitively, it is challenging to predict $z$ (many) given the target $y$ (one). Therefore, the training objective function with the target input can be given as

$$\mathcal{L}_{em-ctc}(\phi) = -\log \sum_{y=\mathcal{B}(z)} P(z|x,y;\phi). \tag{3}$$

The target input is used as guidance to provide the target-side information and reduce the candidates of the latent variable.

**Distillation Loss.** EM-Network is trained in a self-distillation setup where the prediction given source and target inputs serves as a soft label for training the original sequence model. The distillation loss in the proposed method is as follows:

$$\mathcal{L}_{kd-ctc} = \|\hat{z}_{em} - \hat{z}_{org}\|_2^2 \tag{4}$$

where $\hat{z}_{em}$ and $\hat{z}_{org}$ are softmax outputs of the EM-Network and the sequence model, respectively. Note that the argmax value of $\hat{z}$ is the predicted CTC alignment. Since the conventional distillation loss using CE or Kullback-Leibler (KL)-divergence generally fails to converge in the CTC framework, as reported in previous studies (Senior et al., 2015; Takashima et al., 2018; 2019; Yoon et al., 2021b), we follow the distillation loss of Yoon et al. (2021b) that adopts $l_2$ loss function for transferring the latent alignment.

## 2.4 EM-NETWORK FOR AED

**Standard CE Objective.** Different from the CTC, the AED model does not explicitly define the latent alignment $z$ and the many-to-one mapping $\mathcal{B}$. The CE loss for AED can be formulated as

$$\mathcal{L}_{org-ce}(\theta) = -\log P(y|x;\theta). \tag{5}$$

Since simply taking the target $y$ as the AED model's input may cause an obvious but trivial solution where the model converges with the conditional probability $P(y|x,y;\theta) = \delta(y)$, it is difficult to learn a meaningful representation of the target input $y$.

**Modeling Conditional Probability with Target Input.** To address this issue, we employ a masking strategy in the MLM task. The masked version of the target $\tilde{y}$ is applied as the additional input instead of directly using the target $y$. Given the target sequence $y$, we use a masking function $\mathcal{M}$, which randomly masks the tokens of $y$ with the probability of $\lambda$. Then, the masked target $\mathcal{M}(y, \lambda) = \tilde{y}$ is fed to the EM-Network as the auxiliary input. The training objective function with the masked target $\tilde{y}$ can be calculated as follows:

$$\mathcal{L}_{em-ce}(\phi) = -\log P(y|x, \tilde{y}; \phi) \quad \text{where } \tilde{y} = \mathcal{M}(y, \lambda). \tag{6}$$

**Distillation Loss.** In the case of the CE objective function, we utilize the conventional distillation loss, where the KL divergence is computed using softmax outputs between the student and the teacher. The distillation loss for AED-based EM-Network is given by

$$\mathcal{L}_{kd-ce}(\phi, \theta) = D_{KL}(\hat{y}_{em}\|\hat{y}_{org}) \tag{7}$$

where $\hat{y}_{em}$, $\hat{y}_{org}$ denote the softmax outputs of the EM-Network and the sequence model. $D_{KL}$ represents the KL-divergence.

## 3    CONNECTION TO EM ALGORITHM

In this section, we explore the connection with the EM algorithm. The detailed derivations are provided in Appendix B.

**Standard EM Algorithm.** The traditional EM algorithm finds maximum likelihood parameters of the model that depends on unobserved latent variables $z$. It iteratively performs an expectation (E) step and a Maximization step (M step). The E step defines a Q-function $Q(\theta|\theta^{(t)})$ as the expected value of the log-likelihood of $\theta$, which can be formulated as follows:

$$Q(\theta|\theta^{(t)}) = \mathbb{E}_{z|x,\theta^{(t)}}[\log P(x, z; \theta)].$$

Then, the M step computes the parameters that maximize the Q-function found on the E step.

**EM-like Perspective for CTC-based EM-Network** The proposed framework is motivated by the EM algorithm. In the case of the EM-Network for CTC, the Q-function is calculated as follows:

$$Q(\theta|\theta^{(t)}, \rho^{(t)}) = \mathbb{E}_{z|x,y,\theta^{(t)},\rho^{(t)}}[\log P(y, z|x; \theta)]$$
$$= -D_{KL}(P(z|x, y; \theta^{(t)}, \rho^{(t)})\|P(z|x; \theta)) \tag{8}$$
$$\approx -\mathcal{L}_{kd-ctc}(\phi^{(t)}, \theta) \tag{9}$$

where $P(z|x, y; \theta^{(t)}, \rho^{(t)})$ represents the conditional probability of the CTC-based EM-Network in Eq. (3), and $\log P(z|x; \theta)$ denotes the CTC-based sequence model in Eq. (2). Here, we ignore the constant factor $-H(P(z|x, y; \theta^{(t)}, \rho^{(t)}))$ in the fourth equality and the first approximation. As shown in Eq. (8), a negative value of KL-divergence between the EM-Network and the sequence model serves as the Q-function of the EM-Network. However, as aforementioned in Section 2.3, CTC model often fails to converge with the distillation loss using the KL-divergence due to its alignment-free property (Senior et al., 2015; Takashima et al., 2018; 2019; Yoon et al., 2021b). To sidestep this convergence problem, we use $l_2$ loss instead of the KL-divergence, corresponding to the distillation loss $\mathcal{L}_{kd-ctc}$. Eq. (9) shows that $\mathcal{L}_{kd-ctc}$ can be considered as the approximation of the EM-Network's Q-function. Note that $\mathcal{B}(z) = y$ and $\phi = \theta + \rho$ in the CTC-based EM-Network.

Based on the above perspective, we can actually show our proposed training objective in Eq. (4) is a lower bound for the log-likelihood of the original sequence model. Our goal is to maximize the log-likelihood below,

$$\log P(y|x; \theta) \geq \sum_{y=\mathcal{B}(z)} P(z|x, y; \theta^{(t)}, \rho^{(t)}) \log \frac{P(z|x; \theta)}{P(z|x, y; \theta^{(t)}, \rho^{(t)})}$$
$$= -D_{KL}(P(z|x, y; \theta^{(t)}, \rho^{(t)})\|P(z|x; \theta))$$
$$\approx -\mathcal{L}_{kd-ctc}(\phi^{(t)}, \theta) \tag{10}$$

where the first inequality follows from Jensen's inequality. From Eq. (10), we can confirm that a negative value of $\mathcal{L}_{kd-ctc}$ serves as the lower bound for the log-likelihood of the original CTC-based sequence model $\log P(y|x; \theta)$. Therefore, both Eq. (9) and Eq. (10) indicate that maximizing the EM-Network's Q-function is equivalent to maximizing the lower bound of the sequence model's likelihood. The tight lower bound will be additionally discussed in Appendix E.

Maximizing the lower bound (minimizing $\mathcal{L}_{kd-ctc}$) is also closely related to the upper bound for the sequence model's log-likelihood. The conventional KD generally assumes that the teacher's performance determines the upper bound of the student (Zhang et al., 2020; Mishra & Marr, 2018; Clark et al., 2019). Under the assumption, the log-likelihood of the EM-Network (teacher mode) can be regarded as the upper bound for the log-likelihood of the sequence model. Considering that the EM-Network uses the ground truth as the additional input to perform the learning task, this assumption seems more reasonable. Different from the previous offline KD methods where the teacher model is generally fixed, we update the EM-Network and sequence model simultaneously. The training loss curves are given in Figure 4. The proposed distillation loss $\mathcal{L}_{kd-ctc}$ regularizes the EM-Network (teacher mode) to learn the knowledge of the sequence model (student mode), providing the tight upper bound for the sequence model. For the counterpart, minimizing $\mathcal{L}_{kd-ctc}$ also corresponds to making the sequence model mimic the behavior of the EM-Network, indicating that the log-likelihood of the sequence model is close to its upper bound. Therefore, we can derive the tight upper bound by maximizing the lower bound. On top of that, the upper bound can be maximized by minimizing the loss $\mathcal{L}_{em-ctc}$ in Eq. (3), indicating that the proposed objective function partially maximizes the likelihood of the sequence model.

**EM-like Perspective for AED-based EM-Network** When considering the AED, the Q-function can be computed with a similar perspective in Eq. (8), which is given by

$$
\begin{aligned}
Q(\theta|\theta^{(t)}, \rho^{(t)}) &= \mathbb{E}_{y|x,\tilde{y},\theta^{(t)},\rho^{(t)}}[\log P(\tilde{y}, y|x; \theta)] \\
&= -D_{KL}(P(y|x, \tilde{y}; \theta^{(t)}, \rho^{(t)}) \| P(y|x; \theta)) \\
&= -\mathcal{L}_{kd-ce}(\phi^{(t)}, \theta).
\end{aligned}
\tag{11}
$$

Then, we start from the log-likelihood of the AED model, which is formulated in Eq. (5):

$$
\begin{aligned}
\log P(y|x; \theta) &\geq \sum_{\tilde{y}=\mathcal{M}(y,\lambda)} P(y|x, \tilde{y}; \theta^{(t)}, \rho^{(t)}) \log \frac{P(y|x; \theta)}{P(y|x, \tilde{y}; \theta^{(t)}, \rho^{(t)})} \\
&= -D_{KL}(P(y|x, \tilde{y}; \theta^{(t)}, \rho^{(t)}) \| P(y|x; \theta)) \\
&= -\mathcal{L}_{kd-ce}(\phi^{(t)}, \theta)
\end{aligned}
\tag{12}
$$

Eq. (12) indicates that the KL-divergence between the EM-Network ($P(y|x, \tilde{y}; \theta, phi)$) and the AED model ($P(y|x; \theta)$)) accords with the distillation loss in Eq. (7). By Eq. (11), we know that a negative value of the KL-divergence between the EM-Network and the sequence model serves as the Q-function of the EM-Network. Therefore, maximizing the AED-based EM-Network's Q-function is equivalent to maximizing the lower bound of the sequence model's likelihood. As mentioned above, the log-likelihood of the EM-Network serves as the tight upper bound for the sequence model.

## 4 EXPERIMENTAL SETUP

**Speech Recognition.** Our experiments were conducted using the fairseq (Ott et al., 2019) toolkit. We utilized a widely used LibriSpeech (Panayotov et al., 2015) dataset. In the proposed framework, we first pre-trained the sequence model ($\theta$) on the full 960 hours of LibriSpeech (LS-960) by using the objective of the data2vec since it is the current SOTA model in the SSL literature. Then, the whole EM-Network ($\phi$) was fine-tuned with LS-960, where the dev-other subset was used as the validation. The architecture of the sequence model followed the Base setup of the conventional SSL-based model, consisting of 12 transformer blocks. The tunable parameter $\alpha$ was experimentally set to 2. When applying the language model (LM), we used the official 4-gram KenLM (Heafield, 2011). The detailed implementation is additionally presented in Appendix C.

**Machine Translation.** We evaluated the EM-Network on IWSLT'14 and WMT'14 datasets for English-to-German (En-De) and German-to-English (De-En) translation tasks. The implementation

Table 1: (**Speech recognition**) Word error rate (%) on LibriSpeech test set. Bold represents superior results. The results of ILS-SSL and HuBERT were implemented using the public fairseq (Ott et al., 2019) toolkit. For data2vec and wav2vec 2.0 models, we evaluated the performance using the checkpoints provided by the fairseq.

| Method | No LM | | w/ 4-gram LM | |
|---|---|---|---|---|
| | clean | other | clean | other |
| HuBERT Base | 3.79 | 9.04 | 2.68 | 6.21 |
| wav2vec 2.0 Base | 3.40 | 8.42 | 2.59 | 6.19 |
| ILS-SSL Base | 3.44 | 7.79 | 2.65 | 5.84 |
| data2vec Base | 2.78 | 7.02 | 2.43 | 5.56 |
| **Ours, EM-Network** | **2.66** | **6.72** | **2.37** | **5.43** |

Table 2: (**Machine translation**) Comparison of our EM-Network and most recent existing methods on IWSLT'14 and WMT'14 datasets. Bold represents superior results. We reimplemented the results of BiBERT using the public code (Xu et al., 2021), which are shown inside the parentheses.

(a) BLEU score on IWSLT'14 test set.

| Method | En-De | De-En |
|---|---|---|
| Adversarial MLE | - | 35.18 |
| DynamicConv | - | 35.20 |
| Macaron Net | - | 35.40 |
| BERT-Fuse | 30.45 | 36.11 |
| MAT | - | 36.22 |
| Mixed Representations | 29.93 | 36.41 |
| UniDrop | 29.99 | 36.88 |
| BiBERT | 30.45 | 38.61 |
| | (30.48) | (38.66) |
| **Ours, EM-Network** | **31.80** | **39.49** |

(b) BLEU score on WMT'14 newstest2014 test set.

| Method | En-De | De-En |
|---|---|---|
| Large Batch Training | 29.3 | - |
| Evolved Transformer | 29.8 | - |
| BERT Init. (12 layers) | 30.6 | 33.6 |
| BERT-Fuse | 30.75 | - |
| BIBERT | 31.26 | 34.94 |
| | (30.80) | (34.53) |
| **Ours, EM-Network** | **31.30** | **35.40** |

of the EM-Network was based on the official source code provided by the previous work (Xu et al., 2021). Firstly, we pre-trained the sequence model with SSL, and then the whole EM-Network was fine-tuned on the MT task. In the case of the pre-training, the sequence model was trained on the same English and German texts as BiBERT (Xu et al., 2021), derived from the OSCAR (Suárez et al., 2020) corpus. The architecture sequence model was based on the BiBERT, a recent SOTA method on the MT task, and we followed its pre-training scheme. During the fine-tuning, the parameter $\alpha$ and the masking ratio $\lambda$ were set to 2 and 50 %, respectively.

## 5 MAIN EMPIRICAL RESULTS

Since the current SOTA models were typically based on the SSL, we used the pre-trained SSL model in our experiments. Due to the space limit, the experimental results with the fully supervised learning-based models were described in Appendix D. From the results, we observed that the proposed method achieved considerable performance improvement for fully supervised learning.

**Speech Recognition.** We compared the EM-Network with previous SOTA works from the literature, including wav2vec 2.0 (Baevski et al., 2020), HuBERT (Hsu et al., 2021), ILS-SSL (Wang et al., 2022), and data2vec (Baevski et al., 2022). As shown in Table 1, EM-Network outperformed the recent approaches on the LibriSpeech. It is important to note that our approach followed the pre-training scheme of the data2vec, and we applied the proposed scheme to the data2vec's fine-tuning stage. On the LibriSpeech, we improved upon the best data2vec by 0.12 %/0.3 % on the clean/other test datasets with greedy decoding, yielding a relative error rate reduction (RERR) of 4.32 %/4.27 % compared to the data2vec baseline. Also, the proposed approach consistently achieved superior WER results when applying beam-search decoding with the LM. Specifically, compared to the beam-search decoding case, the performance improvement of the EM-Network was much more significant with the greedy decoding. Since the EM-Network considered linguistic (text) information by using the additional target input, the sequence model could produce strong performance even without the LM.

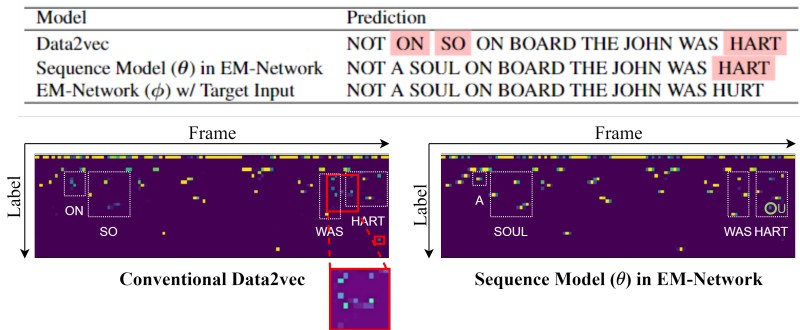

Figure 3: Total frame-wise softmax output examples in test-other dataset, where the target reference is "NOT A SOUL ON BOARD THE JOHN WAS HURT". The x-axis refers to acoustic frames, and the y-axis refers to the character labels. The first label index denotes the "blank" label.

**Machine Translation.**   Table 2a shows a comparison of the proposed method with the recent approaches on IWSLT'14 En-De and De-En translations. We chose the eight best-performed MT algorithms to date: 1) Adversarial MLE (Wang et al., 2019), 2) DynamicConv (Wu et al., 2019) 3) Macaron Net (Lu et al., 2020), 4) BERT-Fuse (Zhu et al., 2020), 5) multi-branch encoders (MAT) (Fan et al., 2020), 6) mixed representations from different tokenizers (Mixed Representation) (Wu et al., 2020), 7) uniting different dropout techniques (UniDrop) (Wu et al., 2021), and 8) BiBERT (Xu et al., 2021). From the results, it is verified that EM-Network outperformed all of them and yielded around 1.4 (En-De) and 0.9 (De-En) BLEU point gains over the previous SOTA results. Recall that EM-Network used the same settings as BiBERT — the main difference was applying the auxiliary network with the additional input $\tilde{y}$ and the proposed self-distillation framework during the fine-tuning stage — yet the EM-Network yielded a significant improvement over BiBERT.

For the high-resource scenario, we conducted experiments on WMT'14 En-De and De-En translations and compared the EM-Network with prior existing works that achieved high BLEU scores on WMT'14 dataset, including large batch training (Ott et al., 2018), Evolved Transformer (So et al., 2019), initializing the BERT by leveraging pre-trained checkpoints (Rothe et al., 2020), BERT-Fuse (Zhu et al., 2020), and BiBERT (Xu et al., 2021). From Table 2b, we verified that our model also gave the best BLEU scores on the high-resource WMT'14 En-De and De-En translations.

# 6 ANALYSIS

**CTC Alignment.**   As visualized in Figure 3, we contrasted the total frame-wise softmax outputs of the best baseline data2vec and our sequence model $\theta$ in the EM-Network. The argmax value of the frame-wise label probability corresponds to the predicted CTC alignment. In Figure 3, the conventional data2vec converted a given speech into "NOT ON SO ON BOARD THE JOHN WAS HART" and made erroneous predictions with "ON SO" and "HART". When considering only the acoustic feature (speech voice), it is challenging to distinguish "ON SO"/"A SOUL" and "HART"/"HURT". However, the proposed sequence model provided a more accurate prediction than the data2vec. Different from the previous seq2seq approaches, the EM-Network performed a fusion of the acoustic (speech source input) and linguistic (text target input) representations. Since the EM-Network could consider not only the acoustic information but also the linguistic one, the sequence model learned better alignment and obtained satisfactory performance. Even though there was a single erroneous word "HART", we can discover the probability regarding "U" (green circle in Figure 3) in the prediction of the sequence model. Also, unlike the data2vec that included many irrelevant label probabilities (red boxes in Figure 3), our sequence model had relatively fewer redundant ones, implying that the use of the target input effectively reduced the candidates of the latent CTC alignment $z$. From the results, it is verified that the proposed framework could learn the promising CTC alignment by using the target input and the self-distillation setup.

**Translation Consistency.**   The token repetition ratio represents the degree of the inconsistency problem in MT (Song et al., 2021; Ghazvininejad et al., 2020). We computed the token repetition ratio by dividing all consecutively repeating tokens by the total number of tokens. The teacher forcing and the greedy decoding were applied to fairly compare the predictions. From the results

Table 3: Token repetition ratio on IWSLT'14 test dataset.

| Model | En-De | De-En |
|---|---|---|
| Reference Test Set | 0.23 | 0.18 |
| BiBERT | 4.76 | 5.62 |
| Sequence Model ($\theta$) in EM-Network | 4.57 | 4.88 |
| EM-Network ($\phi$) w/ Target Input | 4.42 | 4.70 |

in Table 3, we can confirm that the EM-Network effectively addressed the token repetition issue compared to the prior SOTA approach. In the case of the conventional BiBERT, it only considered the past target information during the training via the teacher forcing technique. However, the EM-Network leveraged a more global target context by using the target input $\tilde{y}$. Since our approach used both past and future target information in generating the prediction, it alleviated translation inconsistency and produced better quality. In other words, the sequence model in the proposed framework could benefit from a more optimal and consistent representation of the EM-Network.

**Training Loss Curves.** Figure 4 shows the training loss curves of the EM-Network for the ASR task. We observed that the loss $\mathcal{L}_{em}$ of the whole EM-Network ($\phi$) consistently decreased, implying that soft labels from the EM-Network were updated in each iteration. In this way, unlike the conventional distillation that the knowledge is generally fixed, the EM-Network transferred the knowledge more adaptively. Compared to the previous self-distillation methods (Pei et al., 2021; Kong et al., 2022; Baevski et al., 2022), the proposed distillation framework did not require the additional update strategy, such as a moving-average of param-

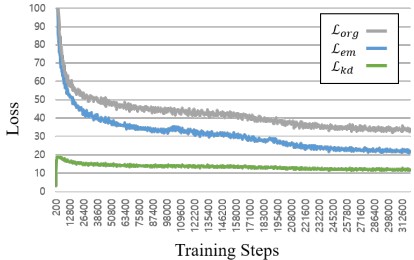

Figure 4: Training loss curves of EM-Network for ASR task.

eters. It could be implemented by simply applying the additional losses $\mathcal{L}_{em}$ and $\mathcal{L}_{kd}$. Also, since $\mathcal{L}_{em}$ was lower than $\mathcal{L}_{org}$ during the training, the prediction of the EM-Network (teacher mode) was more accurate than that of the inner sequence model (student mode). Thus, the sequence model could effectively benefit from the soft labels of the EM-Network.

## 7 RELATED WORKS

**Training with Target Input.** There are many techniques in the seq2seq learning literature that exploit the target-related information. Autoregressive approaches (Vaswani et al., 2017; Chorowski et al., 2015; Graves, 2012) have adopted a teacher forcing technique that supplies a ground truth value as conditional input during the training. The teacher forcing allows the model to predict the current target by utilizing the context representation of the source input and the history of labels. To make good global planning, Feng et al. (2021) proposed the additional seer decoder into the encoder-decoder framework, which embodies future ground truth to guide the behaviors of the conventional decoder. In the case of the non-autoregressive framework, some studies (Higuchi et al., 2020; Chan et al., 2020) attempted to improve the CTC model's prediction by conditioning on previously generated tokens. Also, Yoon et al. (2021a) introduced the Oracle Teacher, which uses the target information to improve the teacher model's performance in the offline distillation setup.

## 8 CONCLUSIONS

In this paper, we introduced EM-Network that can effectively yield a promising latent variable from both source and target inputs. Instead of merely considering the ground truth as the sole target in training, the target was utilized as the model's additional training input. The proposed self-distillation framework enabled the sequence model to benefit from the EM-Network's latent variable in a one-stage manner. We theoretically showed that our training objective, which was motivated by the EM-like algorithm, could be a lower bound for the log-likelihood of the sequence model. Empirically, EM-Network significantly advanced the current state-of-the-art approaches on ASR and MT tasks. We hope our study will draw more attention from the community toward a richer view of the latent variable for seq2seq learning.

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

## A    SOURCE CODE

Source code can be found at `https://github.com/em-net/em-network-anonymous.git`.

## B    THEORETICAL ANALYSIS

**Derivation of Eq. (8)**

$$
\begin{aligned}
Q(\theta|\theta^{(t)}, \rho^{(t)}) &= \mathbb{E}_{z|x,y,\theta^{(t)},\rho^{(t)}}[\log P(y, z|x; \theta)] \\
&= \mathbb{E}_{z|x,y,\theta^{(t)},\rho^{(t)}}[\log[\delta(y - \mathcal{B}(z))P(z|x; \theta)]] \\
&= \sum_{y=\mathcal{B}(z)} P(z|x, y; \theta^{(t)}, \rho^{(t)}) \log P(z|x; \theta) \\
&= -D_{KL}(P(z|x, y; \theta^{(t)}, \rho^{(t)})\|P(z|x; \theta)) \\
&\approx -\mathcal{L}_{kd-ctc}(\phi^{(t)}, \theta)
\end{aligned}
$$

**Derivation of Eq. (10)**

$$
\begin{aligned}
\log P(y|x; \theta) &= \log \sum_z P(y, z|x; \theta) \\
&= \log \sum_z P(y, z|x; \theta)\frac{P(z|x, y; \theta^{(t)}, \rho^{(t)})}{P(z|x, y; \theta^{(t)}, \rho^{(t)})} \\
&= \log \sum_z \delta(y - \mathcal{B}(z))P(z|x; \theta)\frac{P(z|x, y; \theta^{(t)}, \rho^{(t)})}{P(z|x, y; \theta^{(t)}, \rho^{(t)})} \\
&= \log \sum_{y=\mathcal{B}(z)} P(z|x; \theta)\frac{P(z|x, y; \theta^{(t)}, \rho^{(t)})}{P(z|x, y; \phi)} \\
&= \log \sum_{y=\mathcal{B}(z)} P(z|x, y; \theta^{(t)}, \rho^{(t)})\frac{P(z|x; \theta)}{P(z|x, y; \theta^{(t)}, \rho^{(t)})} \\
&\geq \sum_{y=\mathcal{B}(z)} P(z|x, y; \theta^{(t)}, \rho^{(t)}) \log \frac{P(z|x; \theta)}{P(z|x, y; \theta^{(t)}, \rho^{(t)})} \\
&= -D_{KL}(P(z|x, y; \theta^{(t)}, \rho^{(t)}), P(z|x; \theta)) \\
&\approx -\mathcal{L}_{kd-ctc}(\phi^{(t)}, \theta)
\end{aligned}
$$

**Derivation of Eq. (12)**

$$\log P(y|x;\theta) = \log \sum_{\tilde{y}} P(y,\tilde{y}|x;\theta)$$

$$= \log \sum_{\tilde{y}} P(y,\tilde{y}|x;\theta) \frac{P(y|x,\tilde{y};\theta^{(t)},\rho^{(t)})}{P(y|x,\tilde{y};\theta^{(t)},\rho^{(t)})}$$

$$= \log \sum_{\tilde{y}} \delta(\tilde{y} - \mathcal{M}(y,\lambda)) P(y|x;\theta) \frac{P(y|x,\tilde{y};\theta^{(t)},\rho^{(t)})}{P(y|x,\tilde{y};\theta^{(t)},\rho^{(t)})}$$

$$= \log \sum_{\tilde{y}=\mathcal{M}(y,\lambda)} P(y|x;\theta) \frac{P(y|x,\tilde{y};\theta^{(t)},\rho^{(t)})}{P(y|x,\tilde{y};\theta^{(t)},\rho^{(t)})}$$

$$= \log \sum_{\tilde{y}=\mathcal{M}(y,\lambda)} P(y|x,\tilde{y};\theta^{(t)},\rho^{(t)}) \frac{P(y|x;\theta)}{P(y|x,\tilde{y};\theta^{(t)},\rho^{(t)})}$$

$$\geq \sum_{\tilde{y}=\mathcal{M}(y,\lambda)} P(y|x,\tilde{y};\theta^{(t)},\rho^{(t)}) \log \frac{P(y|x;\theta)}{P(y|x,\tilde{y};\theta^{(t)},\rho^{(t)})}$$

$$= -D_{KL}(P(y|x,\tilde{y};\theta^{(t)},\rho^{(t)}) \| P(y|x;\theta))$$

$$= -\mathcal{L}_{kd-ce}(\phi^{(t)},\theta)$$

## C  IMPLEMENTATION DETAILS

Table 4: The number of parameters for each model.

| Task | Method | # of Param. |
|------|--------|-------------|
| ASR | data2vec Base | 93 M |
|     | EM-Network | 103 M ($\theta$: 93 M, $\rho$: 10 M) |
| MT (IWSLT'14) | BiBERT | 206 M |
|     | EM-Network | 225 M ($\theta$: 206 M, $\rho$: 19 M) |
| MT (WMT'14) | BiBERT | 324 M |
|     | EM-Network | 374 M ($\theta$: 324 M, $\rho$: 50 M) |

**Speech Recognition.** For the pre-training stage, we followed the pre-training regime of data2vec (Baevski et al., 2022). After the pre-training, pre-trained models were fine-tuned for the ASR task by applying a linear projection layer. We fine-tuned the model with Adam (Kingma & Ba, 2015) optimizer with an initial learning rate of 1e-4. For the LibriSpeech, the character set had a total of 29 labels plus a word boundary token. The BASE setup contained 12 transformer blocks with model dimension 768 and 8 attention heads. Also, BASE models utilized a batch size of 3.2m samples per GPU and were fine-tuned on eight Quadro RTX 8000 GPUs (each with 48GB of memory), where training updates were set to 320k. Following the recommendation of the language model settings (Baevski et al., 2020), test performance was measured with beam 1,500 for the 4-gram LM. In the case of EM-Network, the architecture of the auxiliary network was based on the self and cross-attention modules, and we additionally employed the feed-forward layers, layer normalization layers, and residual connections for the attention layer.

**Machine Translation.** For the IWSLT'14 dataset, we used four Titan V GPUs (each with 12GB of memory) with 2048 tokens per GPU and accumulated the gradient 4 times. The optimizer was Adam (Kingma & Ba, 2015), with a learning rate of 0.0004. We followed the same byte-pair encoding (BPE) settings of the previous study (Xu et al., 2021). At inference time, we applied beam search with width 4 and a length penalty of 0.6. In the case of the WMT'14 dataset, newstest2012 and newstest2013 were combined as the validation set, and we used newstest2014 as the test set. Followed by the BiBERT(Xu et al., 2021), we used a unified 52K vocabulary for the decoder. We

utilized four Quadro RTX 8000 GPUs (each with 48GB of memory) with a batch size of 4096 tokens per GPU. The gradient was accumulated 32 times. The initial learning rate was set to 0.001.

**Number of Parameter.**   Table 4 compares the number of parameters between the best seq2seq baseline and the EM-Network. We used the best baseline as the inner sequence model of the EM-Network, and thus the parameter for $\theta$ corresponded to that of the best baseline, such as data2vec and BiBERT. The size of the auxiliary network $\rho$ was mainly determined by the vocabulary size. Since the EM-Network for WMT'14 dataset used about 52K vocabulary for the decoder, it required the relatively large parameters for $\rho$. However, during the inference, *only the sequence model was used to generate the prediction*. Since the auxiliary network was removed during the inference, the additional computational load of $\rho$ was only required for the training procedure. Considering the performance improvements of the proposed approach, this computational load for training seems reasonable.

# D   FURTHER EXTENSION: EXPERIMENTAL RESULTS WITH FULLY SUPERVISED LEARNING-BASED MODEL

## D.1   SETUP

For the fully supervised learning-based models, ASR models were implemented in the NeMo (Kuchaiev et al., 2019) toolkit. We trained the EM-Network with LS-960, and the sequence model ($\theta$) was based on a conformer-CTC architecture consisting of 16 conformer (Gulati et al., 2020) blocks with 176 dimensions. We used four Quadro RTX 8000 GPUs (each with 48GB of memory), and 100 epochs were spent for training the models. AdamW algorithm (Loshchilov & Hutter, 2019) was employed as an optimizer with an initial learning rate of 5.0. In the case of Conformer-CTC large, the current SOTA ASR model, we used the pre-trained checkpoint provided by the NeMo (Kuchaiev et al., 2019) toolkit.   For MT task, we evalutated the models with IWSLT'14 datasets for En-De and De-En translations. The sequence model ($\theta$) was based on Transformer (Vaswani et al., 2017). We used four Titan V GPUs (each with 12GB of memory), and 75 epochs were spent for training. BiBERT (Xu et al., 2021), which is the current SOTA MT model, was adopted as the teacher model for conducting the conventional distillation.

## D.2   ASR RESULTS

| Baseline Model | # of Params. | | | clean | other |
|---|---|---|---|---|---|
| Conformer-CTC Large | 122 M | | | 2.78 | 6.18 |
| Conformer-CTC Small | 13 M | | | 4.87 | 12.05 |
| Method | Additional # of Params. for Teacher | Student | KD | clean | other |
| Baseline | None | | None | 4.87 | 12.05 |
| Guided CTC training | 122 M | Conformer-CTC | Offline | 4.63 | 11.56 |
| SKD | 122 M | Small | Offline | 4.53 | 11.33 |
| **Ours, EM-Network** | 1 M | | **Self** | **4.29** | **10.81** |

Table 5: Comparison of word error rate (%) on LibriSpeech test dataset.

In addition to the previous experiments, we checked whether the EM-Network could improve the performance of the sequence models with fully supervised learning. Table 5 shows the results on the LibriSpeech test set. We applied Guided CTC training (Kurata & Audhkhasi, 2019) and SKD (Yoon et al., 2021b) as the competing KD methods for ASR. The Conformer-CTC Large was adopted as the teacher for Guided CTC training and SKD. The Guided CTC training is the effective KD method for CTC-based ASR model. The student can be guided to align with the frame-level alignment of the teacher by using the guided mask. The SKD is the recent KD method in the ASR task, and we used its distillation loss for transferring the soft labels of the EM-Network (Eq. (4)). The main difference between the SKD and the EM-Network was that the EM-Network was trained with the self-distillation setup with the additional target input, while the conventional SKD was based on offline knowledge distillation. Even though the teacher's knowledge in Guided CTC training and

SKD was fixed during the student model training, the soft label of the EM-Network was updated in each iteration so that our approach could transfer the knowledge more adaptively. As shown in Table 5, both Guided CTC training and SKD required the additional teacher model (122 M parameters) to perform the distillation. However, only 1 M parameters, which corresponded to the auxiliary network in the EM-Network, were required to distill the knowledge in the proposed framework. From the results, it is verified that the EM-Network performed well with the fully-supervised learning-based model. Compared to the conventional KD methods using the SOTA ASR model as the teacher, EM-Network achieved better performance on the LibriSpeech test dataset. We confirmed that the proposed method can significantly improve the performance of the sequence model with the fully supervised learning-based model, yielding 4.29 % (RERR: 11.91 %) and 10.81 % (RERR: 10.29 %) on test-clean and test-other, respectively.

### D.3   MT RESULTS

| Baseline Model | # of Params. | | | En-De | De-En |
|---|---|---|---|---|---|
| BiBERT | 206 M | | | 30.48 | 38.66 |
| Transformer | 120 M | | | 27.87 | 34.18 |
| Method | Additional # of Params. for Teacher | Student | KD | En-De | De-En |
| Baseline | None | | None | 27.87 | 34.18 |
| Sequence-level KD | 206 M | Transformer | Offline | 29.33 | 35.97 |
| R-Drop | 0 M | | Self | 29.43 | 35.99 |
| **Ours, EM-Network** | 19 M | | **Self** | **29.84** | **36.47** |

Table 6: Comparison of BLEU score on IWSLT'14 test set.

For MT task, we compared the EM-Network with other previous methods, including sequence-level KD (Kim & Rush, 2016) and R-Drop (Liang et al., 2021), as shown in Table 6. The sequence-level KD is a widely-used KD method in sequence generation tasks, and the R-Drop is a recent and effective regularization method built upon dropout. As the teacher model for the sequence-level KD, we adopted BiBERT Xu et al. (2021), which is a SOTA approach for MT task. The results in Table 6 report that the proposed EM-Network yielded the best BLEU performance on IWSLT'14 dataset, yielding 29.84 and 36.47 on En-De and De-En translation tasks, respectively.

The main difference from the R-Drop was the training loss $\mathcal{L}_{em}$ of the EM-Network. While the R-Drop only consumed the source input, the proposed method utilized the target input when training the EM-Network so that the prediction of the EM-Network (teacher mode) was more accurate than that of the inner sequence model (student mode). From the results, we can verify that the target input was effectively used as guidance to provide the target-side information while producing a considerable performance gain. Compared to the R-Drop that did not require the additional parameters, 19 M parameters were used as the auxiliary network $\rho$ to train the EM-Network. However, since the auxiliary network was removed during the inference, the additional computational load of $\rho$ was only required for the training procedure. Considering the performance improvements, this additional load during the training seemed reasonable. Also, when we reduced the vocabulary size, the additional parameters would be much smaller.

## E   ADDITIONAL VISUALIZATION

We additionally compared the total frame-wise softmax outputs of the best baseline data2vec, our sequence model $\theta$ (student mode) in the EM-Network, and the EM-Network $\phi$ (teacher mode), as visualized in Figure 5. The argmax value of the frame-wise label probability corresponds to the predicted CTC alignment. The conventional data2vec made erroneous predictions with "ON SO" and "HART". When considering only the acoustic feature (speech voice), it is challenging to distinguish "ON SO"/"A SOUL" and "HART"/"HURT". However, the EM-Network gave the correct prediction result with few irrelevant alignments, indicating that the proposed method effectively leveraged the target information to find the optimal CTC alignment. By benefiting from the knowledge of the EM-Network, the proposed sequence model provided a more accurate prediction than the data2vec while it gave "A SOUL" instead of "ON SO".

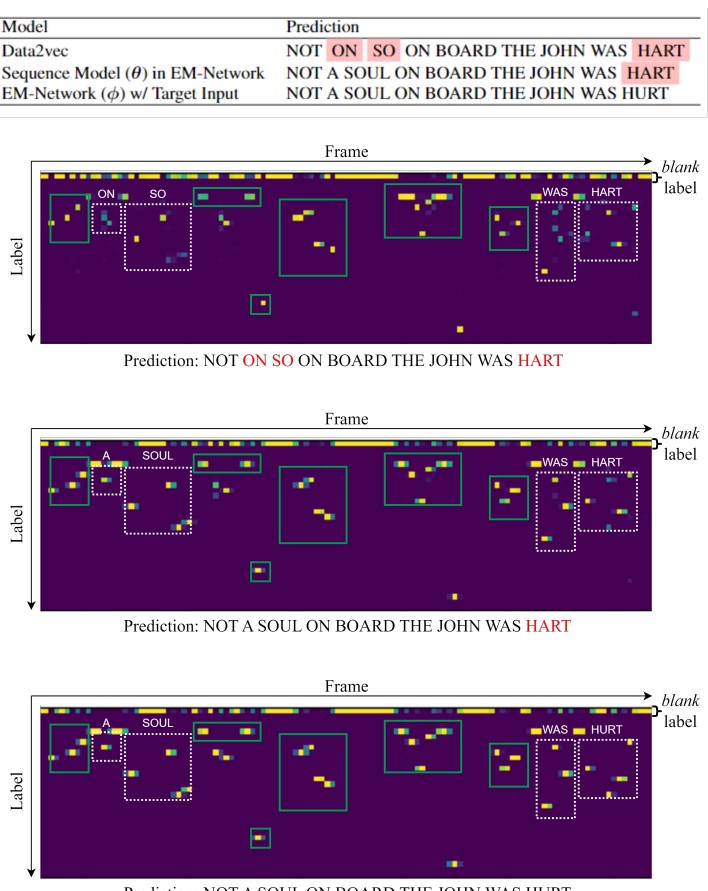

Figure 5: Frame-wise label probability examples for utterance 2609-169640-0020 in LibriSpeech test-other dataset, where the target reference is "NOT A SOUL ON BOARD THE JOHN WAS HURT". Note that its argmax value corresponds to the predicted CTC alignment. The x-axis refers to acoustic frames, and the y-axis refers to the character labels. The first label index represents the "blank" label in the CTC framework.

By Eq. 10, we confirmed that minimizing Eq. 4 maximizes a lower bound of the CTC model's log-likelihood. The lower bound is tight when achieving a low distillation loss between EM-Network and the sequence model, meaning a relatively large overlap between the predictions. Due to the alignment-free property of the CTC framework, the conventional CTC models trained with the same settings, such as model architecture, training data, etc., can have different frame-level alignments, and this difference often leads to the convergence issue of the distillation (Yoon et al., 2021b). However, as shown in Figure 5, both EM-Network (teacher mode) and the sequence model (student mode) could produce similar frame-wise predictions (green boxes in Figure 5), indicating that the proposed method was indeed able to perform the frame-wise distillation effectively and give a considerable alignment overlap, making a tight lower bound.

## F    EFFECT OF $\alpha$ AND $\lambda$

When training the CE-based EM-Network, we considered two tunable parameters $\alpha$ in Eq. (1) and $\lambda$ in Eq. (6). The parameter $\alpha$ was used to balance the distillation loss $\mathcal{L}_{kd}$, and $\lambda$ was applied as the masking probability in the masking function $\mathcal{M}$. We explored the effect of $\alpha$ and $\lambda$ on EM-Network performance, as shown in Figure 6. Firstly, we evaluated the EM-Network on IWSLT'14 En-De translation while varying $\alpha$. $\alpha$ was selected from $\{0, 1, 2, 3, 4\}$, and $\lambda$ was set to 50 %. We observed that on IWSLT'14 En-De, the performance was stably better than the prior SOTA approach, and the

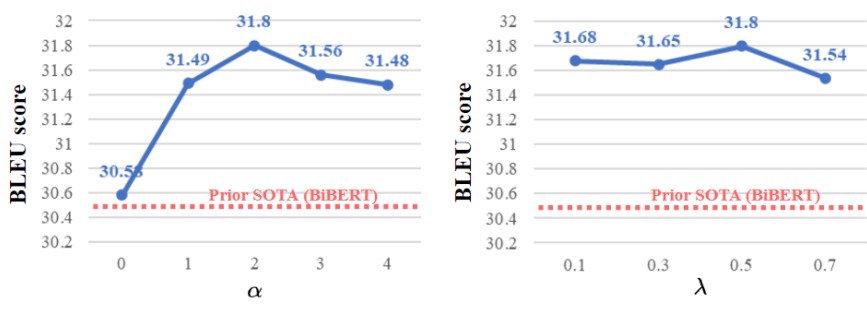

Figure 6: IWSLT'14 En-De performance with varying $\alpha$ and $\lambda$.

best performance was obtained at $\alpha = 2$. Interestingly, the EM-Network with $\alpha = 0$ was slightly better than the performance of the BiBERT, but the difference was negligible. Secondly, we showed the effect of $\lambda$ on EM-Network performance. In this case, $\alpha$ was set to 2, and we selected $\lambda$ from $\{10\%, 30\%, 50\%, 70\%\}$. From the results, it is verified that performance on IWLST'14 En-De achieved the best result when $\lambda = 50\%$.

