# OpenReview forum: "EM-Network: Learning Better Latent Variable for Sequence-to-Sequence Models"
_ICLR.cc/2023/Conference — Submitted to ICLR 2023_

### Official Review · Reviewer_AXAV · 2022-10-24

**Confidence:** 4
**Clarity, Quality, Novelty And Reproducibility:** Please refer to Detailed Comments.
**Correctness:** 2
**Technical Novelty And Significance:** 2
**Empirical Novelty And Significance:** 2
**Recommendation:** 3

**Strength And Weaknesses:**

Strengths:
- This paper designs a novel EM-Network to improve the performance of original sequence-to-sequence models.
- The proposed method achieves new state-of-the-art translation performance on the WMT14 English-German benchmark.

Weaknesses:
- The proposed method is very similar to deep mutual learning. Some strong and similar baselines are missing, such as R-drop, and knowledge distillation from a larger model.
- Actually, the theoretical analysis of the proposed method is the formula derivation of ELBO, which implicitly optimizes the conditional probability of sequence-to-sequence models. Compared with directly optimizing the conditional probabilities of sequence-to-sequence models, the performance gain in this way is unclear to me. I double that the performance gain comes from deep mutual learning.
- The latent variable in this paper is very tricky, i.e., the prediction of the sequence model. It is not sure whether the definition of such latent variables is suitable. There are no detailed discussions with previous latent variable models.


Detailed Comments:

This paper investigates learning a better latent variable for sequence-to-sequence models and achieves new state-of-the-art translation performance on the WMT14 English-German benchmark. Overall, the presentation of this paper should be refined. It is hard for me to understand the motivation of the proposed method for improving sequence learning.

My main concern is the experiment settings. I believe that the proposed knowledge distillation brings performance improvements, but performance gain comes from deep mutual learning. Thus, some strong and similar baselines should be considered, such as R-drop, and knowledge distillation from a larger model, and current experimental results are not convincing enough. In addition, the theoretical analysis in section 3 is the formula derivation of ELBO, which implicitly optimizes the conditional probability of sequence-to-sequence models. Compared with directly optimizing the conditional probabilities of sequence-to-sequence models, the motivation for optimizing ELBO is unclear to me. In my view, the designed latent variable is very tricky, i.e., the prediction of the sequence model. It is not sure whether the definition of such latent variables is suitable. There are no detailed discussions with previous latent variable models.

Questions for the Author(s):
- Please explain the motivation for optimizing ELBO.
- Have you tried other KD methods (e.g., distilling large model) or deep mutual learning (e.g., R-drop or bidirectional agreement on sequence learning)?
- Is it necessary to share parameters between sequence and EM-Network models?
- Have you tried other latent variables beyond the prediction of the sequence model?

Missing References:
- Previous papers on latent variable models, such as latent variable models for NMT.
- Deep mutual learning and R-drop.


**Summary Of The Paper:**

This paper focuses on learning a better latent variable for sequence-to-sequence models. To achieve this goal, the authors design an EM-Network, which first equips an auxiliary network with the sequence model to learn a meaningful representation based on both source and target input, and then distils its knowledge to improve the prediction of the sequence model. This paper connects EM-Network and CTC/CE and provides a theoretical relationship with the standard EM algorithm. Experimental results on ASR/MT tasks show the effectiveness of the proposed method.


**Summary Of The Review:**

This paper investigates learning a better latent variable for sequence-to-sequence models and achieves new state-of-the-art translation performance on the WMT14 English-German benchmark. However, the proposed method is very similar to deep mutual learning, but correspondent comparisons are missing. Experimental results are not convincing enough. In addition, the motivation for optimizing ELBO is unclear to me and there are no detailed discussions with previous latent variable models.

-----------
Thanks for the authors' response and clarification. I appreciate these experiments in Appendix D. However, the improvements over KD/R-drop methods seem marginal. It is better to add significant tests. I tend to keep my score.

---

> ### Author Response · Authors · 2022-11-13
> **First response to reviewer AXAV**
>
> We are very grateful for thorough and detailed comments of the reviewer, which helped improve the manuscript.
>
> We are uploading (a) our point-by-point response to the comments (below) (response to reviewers) and (b) an updated manuscript with blue-colored texts indicating changes.
> ***
> **Concern #1 (1):** The proposed method is very similar to deep mutual learning. Some strong and similar baselines are missing, such as R-drop, and knowledge distillation from a larger model.
>
> **Concern #1 (2):** Have you tried other KD methods (e.g., distilling large model) or deep mutual learning (e.g., R-drop or bidirectional agreement on sequence learning)?
>
> **Concern #1 (3):** Missing References about Deep mutual learning and R-drop.
>
> **Author response:** The main difference from the previous mutual learning approaches [1, 2] is that the proposed method utilizes the target input when training the EM-Network. Since the target input is used as guidance to provide the target-side information, the prediction of the EM-Network (teacher mode) is more accurate than that of the inner sequence model (student mode). Therefore, the sequence model can effectively benefit from the soft labels of the EM-Network, which is additionally discussed in Section 6. The key idea of our framework is to derive a more accurate problem-solving process by referring to the existing solutions so that the EM-Network can provide better guidance to the sequence model, which is a novel technique in the related studies. In addition, we attempt to apply the proposed self-distillation to the CTC framework, an unexplored area in mutual learning research. We have added more descriptions about the difference from the deep mutual learning and the related references in Section1 (pp. 2).
>
> We have added new experimental results, comparing the results with other methods, which can be seen in Appendix D. Since R-Drop used the KL-divergence as the objective, it is hard to apply it to the CTC framework. Therefore, for ASR, we applied recent KD methods, including SKD and Guided CTC training, as the competing approaches. Both SKD and Guided CTC training are the effective KD methods for CTC-based ASR model. From the results, we verified that, compared to the conventional KD methods using the SOTA ASR model as the teacher, EM-Network achieved better performance. In the case of MT, we used R-Drop and Sequence-level KD as the conventional methods. The sequence-level KD is a widely-used KD method in sequence generation tasks, and the R-Drop is a recent and effective regularization method built upon dropout. From the results, we can confirm that the proposed method achieved better performance than other previous approaches. The main difference from the R-Drop was the training loss $\mathcal{L}_{em}$ of the EM-Network. While the R-Drop only consumed the source input, the proposed method utilized the target input when training the EM-Network so that the prediction of the EM-Network (teacher mode) was more accurate than that of the inner sequence model (student mode). From the results, we can verify that the target input was effectively used as guidance to provide the target-side information while producing a considerable performance gain. Thanks for the sincere recommendation.
>
> **Author action:**
>
> (1) We have added more descriptions about the difference from the deep mutual learning in Section1 (pp. 2).
>
> (2) We have added the following references:
>
> - Y. Zhang et al., “Deep mutual learning,” in Proc. CVPR, 2018.
> - X. Liang et al., “R-drop: regularized dropout for neural networks,” in Proc. NIPS, 2021.
>
> (3) We have added new experimental results, which can be seen in Appendix D (pp. 15, 16).
> ***
> **Works cited**
>
> [1] Y. Zhang et al., “Deep mutual learning,” in Proc. CVPR, 2018.
>
> [2] X. Liang et al., “R-drop: regularized dropout for neural networks,” in Proc. NIPS, 2021.

---

> ### Author Response · Authors · 2022-11-13
> **Second response to reviewer AXAV**
>
> **Concern #2 (1):** the theoretical analysis of the proposed method is the formula derivation of ELBO, which implicitly optimizes the conditional probability of sequence-to-sequence models. Compared with directly optimizing the conditional probabilities of sequence-to-sequence models, the performance gain in this way is unclear to me. I double that the performance gain comes from deep mutual learning.
>
> **Concern #2 (2):** Please explain the motivation for optimizing ELBO.
>
> **Author response:** Firstly, we agree that the performance gain of the proposed method comes from a deep mutual learning scheme. We have added more explanations about the performance gain in Section 1 (pp. 2). As mentioned above, different from the conventional deep mutual learning, the proposed method utilizes the target input when training the EM-Network instead of merely considering the ground truth as the sole target in training. Also, we confirmed that our method achieved better performance improvements compared to the R-Drop, as shown in Appendix D.
>
> [About the theoretical analysis] Our main goal is to yield the promising latent variable by leveraging the target sequence as the model's additional training input, and there is a connection with the EM algorithm, which finds maximum likelihood parameters of the model that depends on unobserved latent variables $z$. First, in Eq. (9), we find that the distillation loss (Eq. (4)) between the EM-Network and the sequence model can serve as the Q-function. This can be analogous to the E-step of the EM algorithm.
>
> The proposed self-distillation theoretically means both the Q-function from the EM-like perspective and the lower bound of the sequence model’s log-likelihood. The lower bound is tight when achieving a low distillation loss between EM-Network and the sequence model, indicating a relatively large overlap between the predictions. Regarding the tight lower bound, we have added more descriptions in Appendix E.
>
> Maximizing the lower bound (minimizing $L_{kd-ctc}$) is also closely related to the upper bound for the sequence model's log-likelihood. The conventional KD generally assumes that the teacher's performance determines the upper bound of the student [4, 5, 6]. Under the assumption, the log-likelihood of the EM-Network (teacher mode) can be regarded as the upper bound for the log-likelihood of the sequence model. Considering that the EM-Network uses the ground truth as the additional input to perform the learning task, this assumption seems more reasonable. Different from the previous offline KD methods where the teacher model is generally fixed, we update the EM-Network and sequence model simultaneously, as shown in Figure 4 (training loss curve). The proposed distillation loss  $L_{kd-ctc}$ regularizes the EM-Network (teacher mode) to learn the knowledge of the sequence model (student mode), providing the tight upper bound for the sequence model. For the counterpart, minimizing $L_{kd-ctc}$ also corresponds to making the sequence model mimic the behavior of the EM-Network, indicating that the log-likelihood of the sequence model is close to its upper bound. Therefore, we can derive the tight upper bound by maximizing the lower bound. On top of that, the upper bound can be maximized by minimizing the loss $L_{em-ctc}$ in Eq. (3), indicating that the proposed objective function partially maximizes the likelihood of the sequence model.
>
> Since we allow the target information, such as $y$ and $\tilde{y}$, to be added as the additional input, we believe that this piece of information could largely help avoid the bad optimum problem that often occurs in the distillation for the CTC framework. Also, we did not empirically find any issues with the proposed method converging to an optimal point, as shown in Figure 4. In the conventional self-distillation in image and speech processing [7, 8], the training is unstable, so we need extra models or techniques, such as exponential moving average or stop gradients. However, the proposed algorithm requires no additional technique for stable training due to its tight upper and lower bound.
>
> We acknowledge that detailed explanations were not presented enough. We have added explanations about the theoretical analysis (pp. 6). Thank you for careful reading.
>
> **Author action:**
>
> (1) We have added more explanations about the performance gain in Section 1 (pp. 2).
>
> (2) We have added more descriptions  about the tight lower bound in Appendix E (pp. 17).
>
> (3) We have added explanations about the theoretical analysis (pp. 6).
>
> (4) We have added the following references:
>
> - Z. Zhang et al., “Distilling knowledge from well-informed soft labels for neural relation extraction,” in Proc. AAAI, 2020.
> - A. Mishra and D. Marr, “Apprentice: using knowledge distillation techniques to improve low precision network accuracy,” in Proc. ICLR, 2018.
> - K. Clark et al., “Bam! born-again multi-task networks for natural language understanding,” in Proc. ACL, 2019.

---

> ### Author Response · Authors · 2022-11-13
> **Third response to reviewer AXAV**
>
> **Concern #3 (1):** The latent variable in this paper is very tricky, i.e., the prediction of the sequence model. It is not sure whether the definition of such latent variables is suitable. There are no detailed discussions with previous latent variable models.
>
> **Concern #3 (2):** Missing references: Previous papers on latent variable models, such as latent variable models for NMT.
>
> **Author response:** In Section 1, we present the previous examples of the latent variable in the seq2seq tasks. Firstly, CTC alignment is a popular latent alignment for seq2seq learning. The conventional CTC model, which is alignment ASR model, defines the latent alignment $z$ to learn the mapping between the speech feature $x$ and the word sequence $y$. The argmax value of the total frame-wise softmax output is the predicted CTC alignment, and previous KD studies for CTC generally use this softmax knowledge for conducting KD. Therefore, using CTC alignment as the latent variable seems reasonable.
>
> The conventional CTC models often converge to sub-optimal alignment distributions and produce over-confident predictions [9, 10, 11]. Since there is an exponential number of possible alignment paths, and the alignment information between source and target sequences is rarely available during training, settling on the optimal alignment is quite challenging. However, the proposed EM-Network computes the posterior $P(z|x,y)$ for the loss and can predict the optimal CTC alignment $z$ by leveraging the source and target inputs.
>
> In the case of AED, since there is no explicit latent alignment like CTC, it is challenging to directly apply the same scheme. Inspired by the BERT family models that employ MLM to learn the contextualized representations $z$, the EM-Network for AED aims at learning a better contextualized representation using the masked target $\tilde{y}$ and offering the desired performance on the learning task. The contextualized representation is also a widely used concept in the SSL literature. For the recent NLP studies, including NMT, a large pre-trained LM is highly required to obtain the contextualized representation [12, 13, 14]. The proposed EM-Network provides more robust contextualized representations that can benefit the learning task.
>
> However, we agree that the detailed explanations were not enough. We have added more information about the latent variable in Section 1 (pp. 1, 2). The references related to NMT are also given in Section 1 (pp. 2).
>
> **Author action:** We have added more information about the latent variable in Section 1 (pp. 1, 2) with the references regarding NMT.
> ***
> **Concern #4:** Have you tried other latent variables beyond the prediction of the sequence model?
>
> **Author response:** In our experiments, we only consider the prediction to perform the proposed self-distillation. Firstly, the proposed Q-function and lower bound are derived as the KL-divergence between the EM-Network and the sequence model. When applying other knowledge, such as representation, we need many changes deviating from the original analysis. Secondly, the previous distillation-based studies [1, 2, 3] generally consider the model’s prediction for training. Thank you for careful reading.
> ***
> **Works Cited**
>
> [3] J. W. Yoon, H. Lee, H. Y. Kim, W. I. Cho, and N. S. Kim, “Tutornet: towards flexible knowledge distillation for end-to-end speech
>  recognition,” IEEE/ACM Transactions on Audio, Speech, and Language Processing, 2021.
>
> [4] Z. Zhang et al., “Distilling knowledge from well-informed soft labels for neural relation extraction,” in Proc. AAAI, 2020.
>
> [5] A. Mishra and D. Marr, “Apprentice: using knowledge distillation techniques to improve low precision network accuracy,” in Proc. ICLR, 2018.
>
> [6] K. Clark et al., “Bam! born-again multi-task networks for natural language understanding,” in Proc. ACL, 2019.
>
> [7] A. Baevski, W. Hsu, Q. Xu, A. Babu, J. Gu, and M. Aulim, “Data2vec: a general framework for self-supervised learning in speech, vision and language,” in Proc. ICML, 2022.
>
> [8] X. Chen, and K. He, “Exploring simple siamese representation learning,” in Proc. CVPR, 2021.
>
> [9] H. Liu et al., “Connectionist temporal classification with maximum entropy regularization,” in Proc. NIPS, 2018.
>
> [10] J. Yu et al., “Fastemit: low-latency streaming asr with sequence-level emission regularization,” in Proc. ICASSP, 2021.
>
> [11] Y. Miao et al., “Eesen: End-to-end speech recognition using deep rnn models and wfst-based decoding,” in Proc. ASRU, 2015
>
> [12] H. Xu, B. Durme, and K. Murray, “Bert, mbert, or bibert? a study on contextualized embeddings for neural machine translation,” in Proc. EMNLP, 2021.
>
> [13] Z. Wu, L. Wu, Q. Meng, Y. Xia, S. Xie, T. Qin, X. Dai, and T. Liu, “Unidrop: a simple yet effective technique to improve transformer without extra cost,” in Proc. NAACL, 2021.
>
> [14] J. Zhu, Y. Xia, L. Wu, D. He, T. Qin, W. Zhou, H. Li, and T. Liu, “Incorporating bert into neural machine translation,” in Proc. ICLR, 2020.

---

> ### Author Response · Authors · 2022-11-23
> **Thank you & looking forward to hearing your feedback**
>
> Dear Reviewer AXAV:
>
> Thank you very much for your precious time and valuable comments. We hope our responses have clarified your concerns. Please kindly let us know if you have more questions. We will try our best to answer them before the rebuttal ends.
>
> Best,
>
> Authors

---

> ### Author Response · Authors · 2022-11-24
> **Reply to reviewer AXAV (regarding performance improvements for MT)**
>
> **Concern:** However, the improvements over KD/R-drop methods seem marginal. It is better to add significant tests. I tend to keep my score.
>
> **Author response:** Thanks for your reply, and we are glad to know most of your concerns have been addressed.
>
> From the results in Table 6, we confirmed that our method yielded around 0.41 (En-De) and 0.48 (De-En) BLEU point gains over the R-Drop [1]. The recent UniDrop [2] yielded 0.47 (De-En) gains over the competing method, and also the recent MT study [3] achieved about 0.3 BLEU point gain for De-En translation task compared to the prior SOTA method, indicating that our performance improvements over KD/R-drop methods seem quite significant.
>
> While there was no significant performance difference between R-Drop and KD, which can be seen in Table 6, our method could achieve considerable performance improvements for all configurations. In summary, our proposed approach is clearly better than the other methods, including R-Drop and KD.
>
> Could you please clarify the point about "the significant tests"?
>
> [1] X. Liang et al., “R-drop: regularized dropout for neural networks,” in Proc. NIPS, 2021.
>
> [2] Z. Wu, L. Wu, Q. Meng, Y. Xia, S. Xie, T. Qin, X. Dai, and T. Liu, “Unidrop: a simple yet effective technique to improve transformer without extra cost,” in Proc. NAACL, 2021.
>
> [3] L. Wu, S. Xie, Y. Xia, Y. Fan, J. Lai, T. Qin, and T. Liu, "Sequence generation with mixed representations," in Proc. ICML, 2020.

---

> > ### Comment · Reviewer_AXAV · 2022-12-02
> > **Comment**
> >
> > Thanks for your detailed response and clarification. I appreciate these experiments in Appendix D. However, the improvements over KD/R-drop methods seem marginal (<0.5 BLEU). It is better to add significant tests. You can use the bootstrap re-sampling (Koehn, 2004) method to test the significant difference. I propose that the authors consider comparing the performance of different systems using some more recent, neural metric (e.g., BLEURT and COMET) in addition to BLEU as it would allow drawing better comparisons. I am also confused about the implementation of the basic ASR systems. I remember that the ESPNet toolkit provides an implementation of combining the CTC decoding and seq2seq model, which achieves a better performance than the pure seq2seq model. In addition, I am not satisfied with the choice of latent variable in this paper. Current clarification is not addressed my concern well and I think it is too tricky and not foundational. Thus, I tend to keep my score for the current version.

---

> > > ### Author Response · Authors · 2022-12-06
> > > **Re: Comment (1)**
> > >
> > > **(Main) Concern:** The improvements over KD/R-drop methods seem marginal (<0.5 BLEU).
> > >
> > > **Author response:** Thanks for your reply. To deal with the performance improvement issue, we have updated new experiment results by varying the tunable parameter $\alpha$ in Eq. (1), which was used to balance the distillation loss. While $\alpha$ was experimentally set to 2 for SSL-based MT models, which can be seen in Appendix F, we found that the fully-supervised MT model achieved the best performance when $\alpha=5$. For competing methods, including Sequence-level KD and R-Drop, we followed configurations of original papers.
> > >
> > > Experiments on IWSLT:
> > >
> > > |Method|Additional # of Params. for Teacher|En-De|De-En|
> > > |:---|:---:|:---:|:---:|
> > > |Baseline|None|27.87|34.18|
> > > |Sequence-level KD|206 M|29.33|35.97|
> > > |R-Drop|0 M|29.43|35.99|
> > > |Ours, EM-Network|1 M|29.94|36.53|
> > >
> > > From the results, we can verify that the proposed method achieved considerable performance improvements over KD/R-drop methods, yielding around 0.51 (En-De) and 0.54 (De-En) BLEU point gains over the competing method. Considering that the recent MT studies [1,2] yielded around 0.3~0.47 gains over the competing method on IWSLT, our performance improvements over KD/R-drop methods seem significant. We will update these experimental results in the revised version. We are very grateful for your insightful feedback, which helped improve the manuscript.
> > >
> > > [1] Z. Wu, L. Wu, Q. Meng, Y. Xia, S. Xie, T. Qin, X. Dai, and T. Liu, “Unidrop: a simple yet effective technique to improve transformer without extra cost,” in Proc. NAACL, 2021.
> > >
> > > [2] L. Wu, S. Xie, Y. Xia, Y. Fan, J. Lai, T. Qin, and T. Liu, "Sequence generation with mixed representations," in Proc. ICML, 2020.
> > >
> > > ---
> > >
> > > **Concern:** I am also confused about the implementation of the basic ASR systems. I remember that the ESPNet toolkit provides an implementation of combining the CTC decoding and seq2seq model, which achieves a better performance than the pure seq2seq model.
> > >
> > > **Author response:** As the reviewer commented, the ESPNET toolkit provides the implementation of the hybrid ctc/attention [1]. However, there are mainly two reasons about the model selection:
> > >
> > > Reason 1) The proposed method aims at predicting a better CTC alignment $z$ by leveraging the source and target inputs. To properly check the effectiveness of the EM-Network in the CTC framework, we used the pure CTC model as the baseline.
> > >
> > > Reason 2) The conformer-CTC, which was used as the baseline in the fully-supervised ASR setting, is a recent ASR model that can achieve powerful performances (+ non-autoregressive nature), and recent ASR studies [2,3] adopt the conformer-CTC as the baseline. Also, we additionally have applied beam search decoding with a strong language model (Transformer LM) to further improve the performance of the ASR baseline.
> > >
> > > |Baseline Model|test-clean|test-other|
> > > |:---|:---:|:---:|
> > > |Conformer-CTC Large (teacher for Guided CTC training and SKD)|2.01%|4.69%|
> > > |Conformer-CTC Small (student baseline)|2.85%|8.34%|
> > >
> > > |Method|Additional # of Params. for Teacher|test-clean|test-other|
> > > |:---|:---:|:---:|:---:|
> > > |Baseline|None|2.85%|8.34%|
> > > |Guided CTC training|122 M|2.88%|8.27%|
> > > |SKD|122 M|2.82%|8.00%|
> > > |Ours, EM-Network|1 M|2.70%|7.80%|
> > >
> > > From the results, we confirmed that the ASR baseline achieved 2.85% and 8.34% on test-clean and test-other datasets, respectively, with the strong LM. In addition, the proposed method achieved considerable performance improvements with beam-search decoding compared to the other approaches in the fully supervised setting, yielding WER 2.70 % (RERR 5.26 %) on test-clean and WER 7.80 % (RERR 6.47 %) on test-other. The results show that Guided CTC training did not always perform better than the student baseline with the beam-search decoding, meaning that applying LM was more challenging than greedy decoding, as reported in [4]. While the conventional KD frameworks were not sufficiently effective to improve the baseline for such scenario, the EM-Network always improved the performance of the original baseline. We will add more description about the model selection and these experimental results in the revised version. Thank you for your careful reading.
> > >
> > >
> > > [1] S. Watanabe et al., “Hybrid ctc/attention architecture for end-to-end speech recognition,” IEEE Journal of Selected Topics in Signal Processing, 2017.
> > >
> > > [2] S. Kim et al., “Squeezeformer: an efficient transformer for automatic speech recognition,” in Proc. NeurIPS, 2022.
> > >
> > > [3] S. Ding et al., “Audio Lottery: Speech Recognition Made Ultra-Lightweight, Noise-Robust, and Transferable,” in Proc. ICLR, 2022.
> > >
> > > [4] J. W. Yoon et al., “Tutornet: towards flexible knowledge distillation for end-to-end speech recognition,” IEEE/ACM Transactions on Audio, Speech, and Language Processing, 2021.

---

> > > ### Author Response · Authors · 2022-12-06
> > > **Re: Comment (2)**
> > >
> > > **Concern:**  I am not satisfied with the choice of latent variable in this paper. Current clarification is not addressed my concern well and I think it is too tricky and not foundational.
> > >
> > > **Author response:** In our paper, we consider CTC alignment and contextualized representation as latent variables.
> > >
> > > CTC alignment) Firstly, the CTC alignment is a popular latent alignment for seq2seq learning. Considering that most studies regarding CTC, including the original CTC paper, use the concept of the latent alignment to explain the CTC, using CTC alignment as the latent variable seems reasonable.
> > >
> > > Contextualized representation) Inspired by the BERT family models that employ MLM to learn the contextualized representations $z$, the EM-Network for AED aims at learning a better contextualized representation, and the contextualized representation is also a widely used concept in the SSL literature [1-5].
> > >
> > > According to the reviewer’s comments, we already added explanations and references related to the CTC alignment and the contextualized representation.
> > >
> > > In addition to the latent variable, we acknowledge that the proposed method can be viewed from the deep mutual learning perspective. Even though we added the description about the relationship between our work and deep mutual learning in the current manusciprt, we will add more explanations about this point in the revised version.
> > >
> > > [1] H. Xu, B. Durme, and K. Murray, “Bert, mbert, or bibert? a study on contextualized embeddings for neural machine translation,” in Proc. EMNLP, 2021.
> > >
> > > [2] Z. Wu, L. Wu, Q. Meng, Y. Xia, S. Xie, T. Qin, X. Dai, and T. Liu, “Unidrop: a simple yet effective technique to improve transformer without extra cost,” in Proc. NAACL, 2021.
> > >
> > > [3] J. Zhu, Y. Xia, L. Wu, D. He, T. Qin, W. Zhou, H. Li, and T. Liu, “Incorporating bert into neural machine translation,” in Proc. ICLR, 2020.
> > >
> > > [4] W. Hsu, B. Bolte, Y. Tsai, K. Lakhotia, R. Salakhutdinov, and A. Mohamed, "Hubert: self-supervised speech representation learning by masked prediction of hidden units," IEEE/ACM Transactions on Audio, Speech, and Language Processing, 2021.
> > >
> > > [5] A. Chung et al., "W2v-bert: combining contrastive learning and masked language modeling for self-supervised speech pre-training," arXiv preprint arXiv:2108.06209v2, 2021.

---

> > > ### Author Response · Authors · 2022-12-09
> > > **We are anticipating your feedback**
> > >
> > > Dear Reviewer AXAV, since the discussion period will end soon, we would greatly appreciate it if you could take a look at our reply and let us know if you have any remaining questions.
> > >
> > > We look forward to addressing any remaining concerns before the end of the discussion period.

---

> > > ### Author Response · Authors · 2022-12-14
> > > **Dear Reviewer AXAV (regarding performance improvements over KD/R-drop)**
> > >
> > > To deal with the performance improvement issue (**main concern**), we have updated new experiment results.
> > > Also, we have added more descriptions and results regarding the fully-supervised ASR.
> > >
> > > We are wondering if there are any unsolved questions or concerns.
> > >
> > > Many thanks, The authors

---

### Official Review · Reviewer_ngMM · 2022-10-24

**Confidence:** 5
**Correctness:** 3
**Technical Novelty And Significance:** 3
**Empirical Novelty And Significance:** 3
**Recommendation:** 5

**Clarity, Quality, Novelty And Reproducibility:**

Saying that the latent variable in conventional models does not depend on the target y was a bit confusing to me. Because when you actually compute the CTC loss, you actually compute the posterior p(z|x,y) for the loss. When carrying out the forward-backward computation for the error signal, this is actually exactly this posterior which you compute there.

So this is just a new way to compute this posterior. Instead of computing it exactly, it is now approximated by a separate model.
But the paper does not really explain this well and is rather explaining it in a somewhat obscure and confusing way.

For example: "leveraging the target sequence as the model’s additional training input". This is wrong. In training of CTC or so, surely you leverage the target sequence. You do that when computing p(z|x,y) for the CTC loss.

Or: "their shortcoming is that the latent variable captures only the source-side information". I don't really understand this. Surely it must be this way.

Further, I was all the time confused about how this actually applies to the AED model, and only somewhere in the middle realized that for the AED model, it does not really apply at all, but just uses a similar motivated loss and training method including the separate model.
What is actually meant by "conventional sequence model" or "seq2seq learning"? This should really be clarified further. CTC is give as one example. So is it only about CTC? Or what about the other common models like hybrid NN-HMM, Transducer, attention-based encoder-decoder (AED), or segmental models? Later through reading, I think it's about CTC for speech, and AED for translation, so only those two model types, and nothing else. But this should be totally clear from the beginning on in the paper, probably even in the abstract.

What is meant by CE-based objective function? CTC is a CE-based objective function as well. So this statement or categorization does not make sense to me. This is also not really defined anywhere, so this is a critical flaw. From other parts of the paper, I assume that "CE-based objective" is a very strange way of referring to AED based models?

It seems that the CE-based EM-network does not really fit well into the whole mathematical motivation of the paper.
Distillation loss, eq 4: Are these the latent variables? The latent variables are discrete, so the equation does not make sense then. Or are these actually the log probs for the latent vars? Or what exactly are those \hat{z}?

I think it distracts from the actual work to use SSL pretrained models. The SSL part is in no way relevant for the work, and potentially hides some aspects.

"Latent CTC Alignment. As visualized in Figure 3, we contrasted the latent alignments" - what does this mean? The latent variables are discrete. The figure does not seem to show any discrete values. So what exactly do we see in the figure?


**Strength And Weaknesses:**

Strengths:

- Interesting idea to improve the latent variable, or in general for regularization.
- There is the intent to release the source code.

Weaknesses:

- Some important aspects are unclear. See below.
- The whole mathematical formulation has several issues. I think this already starts with framing it as EM-like. I cannot really pinpoint this to one specific aspect, but there are many things which are not quite right, and this also shows that in the end it needs many changes deviating from the original motivation, e.g. using mean-squared-error for the distillation instead of KL, and then also the AED model does not fit into the picture at all, and also requires another masked-based loss for the separate posterior model. This does not mean that the proposed method by itself is bad, but just the mathematical derivation or motivation seems wrong.
- It uses pretrained models and brings in the aspect of SSL, although this is totally irrelevant for the proposed method, and actually distracts from it. This also means it cannot properly be compared to other results from the literature. It rather should follow the original exact task, and only use the right training data, and not use SSL.
- It would be interesting to further study the behavior of the latent variable, specifically the alignments. There is some figure and some discussion on it but this could be extended much more.
- The source code is just a dump of Fairseq, probably including their modifications. But there is no explanation of what code parts are actually changed, and also is not properly forked from the Fairseq repo, so it's really difficult to see the actual differences and changes to Fairseq.


**Summary Of The Paper:**

Tasks: speech recognition, translation

Models: CTC for speech, attention-based encoder-decoder (AED) for translation

The paper proposes a new training method to improve the quality of the latent variable and thus the overall quality of the model. Or in the case of AED, it acts as a sort of regularization.

The method is inspired from expectation-maximization (EM). Conventional EM computes p(z|x,y) given the model p_seq2seq(z|x). In the proposed method, the quantity p(z|x,y) is not calculated from the original model, but there is a separate model for it, p_separate(z|x,y). Then it uses distillation (KL or mean-squared-error) to train p_seq2seq based on this. This separate model for p(z|x,y) shares the same encoder (param sharing) but then has some additional cross-attention to y on top.

Everything is jointly trained with the loss: L_org + L_em + alpha * L_kd

L_org is the original loss: CTC loss or framewise CE for AED.

L_kd is the distillation loss (KL or mean-squared-error). Mean-squared-error is taken when L_org is CTC.

L_em is for training the separate posterior model for p(z|x,y). It's defined as - log p(y|x,y) = - log sum_{z:y} p(z|x,y).

For the AED model, where there is no latent variable, this is changed further to use a masked training loss instead for L_em.


**Summary Of The Review:**

The method by itself is interesting and probably could be a useful building block for better regularization.

I think many aspects need more clarification, and maybe even the whole mathematical formulation and derivation should be improved, or reformulated.

Further, I think the baselines are badly chosen, and distract from studying the actual method.

The analysis could also be extended.

---

> ### Author Response · Authors · 2022-11-13
> **First response to reviewer ngMM**
>
> We are very grateful for thorough and detailed comments of the reviewer, which helped improve the manuscript.
>
> We are uploading (a) our point-by-point response to the comments (below) (response to reviewers) and (b) an updated manuscript with blue-colored texts indicating changes.
> ***
> **Concern #1:** This also shows that in the end it needs many changes deviating from the original motivation, e.g. using mean-squared-error for the distillation instead of KL.
>
> **Author response:** As the reviewer commented, EM-Network for CTC uses mean-squared-error (MSE) for the distillation instead of KL-divergence. This is because, as mentioned in Section 2.3, CTC model fails to converge with the distillation loss using the KL-divergence due to its alignment-free property [1-4]. To sidestep the convergence problem, we follow the distillation loss of [1] that adopts the MSE for transferring the knowledge of the EM-Network. In other words, the MSE is required to perform the distillation in the CTC-framework instead of KL-divergence. We have added more CTC outputs examples in Appendix E and confirmed that the proposed method was indeed able to perform the frame-wise distillation effectively.
>
> The MSE-based distillation loss still follows the original motivation. By using the MSE instead of KL-divergence, the proposed method can effectively transfer the alignment knowledge of the EM-network. Since the MSE is important to apply the EM-Network to the CTC framework, the change regrading MSE seems reasonable.
>
> **Author action:**  We have added more CTC outputs examples in Appendix E (pp. 16, 17).
> ***
> **Concern #2:** the AED model does not fit into the picture at all, and also requires another masked-based loss for the separate posterior model.
>
> **Author response:** [About the picture] Figure 1 and Figure 2 can cover not only the CTC-based EM-Network but also the AED case. When consider the AED model, the sequence model and the target input correspond to the AED and $\tilde{y}$, respectively. However, we acknowledge that detailed explanations are not presented enough. In Figure 1 and Figure 2, we have added pictorial representation to make it easier for the reader to consume the information about AED. Also, we have added the description about the teacher forcing in the caption of Figure 2.
>
> [About mask-based loss for AED] Different from the CTC, there is no explicit latent alignment for AED. Therefore, it is challenging to directly apply the same training scheme as the EM-Network for CTC. Simply taking the target $y$ as the additional input may cause an obvious but trivial solution, where the model converges with the conditional probability $P(y|x,y) = \delta(y)$. The proposed masked-based loss allows the AED model to use the target information for model training.
>
> The main difference between the CTC-based EM-Network and the AED-based one relies on the target input. The AED-based EM-Network apply $\tilde{y}$ as the target input for training while CTC-based EM-Network uses $y$ as the target input. However, this difference is required to apply the proposed framework to the AED model, where there is no explicit latent alignment. By covering both CTC (using latent alignment) and AED (no latent alignment) models, the proposed method can be used for a wide range of seq2seq tasks. Also, we have added the theoretical analysis for AED-based EM-Network, which can be seen in Section 3. We check that the EM-Network for AED can be interpreted by a similar EM-like perspective, like CTC-based EM-Network.
>
> **Author action:**
>
> (1) We have added pictorial representation to make it easier for the reader to consume the information: Figure 1 and Figure 2.
>
> (2) We have added the description about the teacher forcing in the caption of Figure 2.
>
> (3) We have added the theoretical analysis for AED-based EM-Network, which can be seen in Section 3 (pp. 6).
> ***
> **Works Cited**
>
> [1] J. W. Yoon, H. Lee, H. Y. Kim, W. I. Cho, and N. S. Kim, “Tutornet: towards flexible knowledge distillation for end-to-end speech recognition,” IEEE/ACM Transactions on Audio, Speech, and Language Processing, 2021.
>
> [2] A. Senior, H. Sak, F. C. Quitry, T. Sainath, K. Rao, et al., “Acoustic modelling with cd-ctc-smbr lstm rnns,” In Proc. ASRU, 2015.
>
> [3] R. Takashima, S. Li, and H. Kawai, “An investigation of a knowledge distillation method for ctc acoustic models,” In Proc. ICASSP, 2018.
>
> [4] R. Takashima, S. Li, and H. Kawai, “Investigation of sequence-level knowledge distillation methods for ctc acoustic models,” In Proc. ICASSP, 2019.

---

> ### Author Response · Authors · 2022-11-13
> **Second response to reviewer ngMM**
>
> **Concern #3 (1):** It uses pretrained models and brings in the aspect of SSL, although this is totally irrelevant for the proposed method, and actually distracts from it. This also means it cannot properly be compared to other results from the literature. It rather should follow the original exact task, and only use the right training data, and not use SSL.
>
> **Concern #3 (2):** I think it distracts from the actual work to use SSL pretrained models. The SSL part is in no way relevant for the work, and potentially hides some aspects.
>
> **Author response:** Since the current SOTA models are typically based on the SSL, we use the pre-trained SSL model in our experiments. The R-Drop [5], which is a recent mutual learning approach, also took strong pre-trained models as their backbones, and applied their method to the fine-tuning stage. From the SSL-based experimental results, we confirmed that the proposed approach using the pre-trained model achieves SOTA results in ASR and MT tasks.
>
> However, we agree that conducting the experiments with fully supervised learning setting is very helpful for checking the effectiveness of the proposed method and improving the quality of the paper. We have added new experimental results, which can be seen in Appendix D. From the results, we confirmed that the EM-Network could achieved considerable performance improvements with fully supervised learning, compared to the previous studies. Thanks for the sincere recommendation.
>
> **Author action:** We have added new experimental results, which can be seen in Appendix D (pp. 15, 16).
>
> ***
> **Concern #4:** It would be interesting to further study the behavior of the latent variable, specifically the alignments. There is some figure and some discussion on it but this could be extended much more.
>
> **Author action:** We have added more descriptions on the CTC alignment, which can be seen in Appendix E (pp. 16, 17).
> ***
> **Concern #5:** The source code is just a dump of Fairseq, probably including their modifications. But there is no explanation of what code parts are actually changed, and also is not properly forked from the Fairseq repo, so it's really difficult to see the actual differences and changes to Fairseq.
>
> **Author action:** We have added more explanations (about the main differences from original Fairseq) in the github profile.
> ***
>
> **Works Cited**
>
> [5] X. Liang et al., “R-drop: regularized dropout for neural networks,” in Proc. NIPS, 2021.

---

> > ### Comment · Reviewer_ngMM · 2022-11-29
> > **Comment**
> >
> > > Since the current SOTA models are typically based on the SSL
> >
> > This is wrong. The standard ASR task is librispeech *without* additional audio data, and for this task, none of the SOTA models use SSL.

---

> > > ### Author Response · Authors · 2022-11-30
> > > **Reply to reviewer ngMM**
> > >
> > > Thanks for your reply. For only the Librispeech dataset (supervised setting), it is true that the SOTA model is not based on the SSL framework. However, if the SSL-based model is pre-trained by an additional audio dataset without transcribed text and then fine-tuned on the Librispeech dataset, the SSL-based ASR model is the current SOTA for the LibriSpeech benchmark [1, 2, 3, 4].
> > >
> > > Since research interest in SSL models for ASR has grown rapidly, we apply our algorithm not pre-training but for the fine-tuning task of SSL.
> > > Since the data2vec [3], which is the current SOTA SSL model for ASR, experimented with Librispeech 960h for pre-training and fine-tuning, we reported WER results for the fine-tuning task of SSL models.
> > >
> > > We believe our results are significant for those studying SSL for ASR.
> > >
> > > [1] A. Baevski, H. Zhou, A. Mohamed, and A. Michael, "Wav2vec 2.0: a framework for self-supervised learning of speech representations," in Proc. NIPS, 2020.
> > >
> > > [2] C. Chiu et al., "Self-supervised learning with random-projection quantizer for speech recognition," in Proc. ICML, 2022.
> > >
> > > [3] A. Baevski, W. Hsu, Q. Xu, A. Babu, J. Gu, and M. Aulim, "Data2vec: a general framework for self-supervised learning in speech, vision and language," in Proc. ICML, 2022.
> > >
> > > [4] "W2v-bert: combining contrastive learning and masked language modeling for self-supervised speech pre-training," arXiv
> > > preprint arXiv:2108.06209v2, 2021.

---

> > > > ### Comment · Reviewer_ngMM · 2022-11-30
> > > > **Comment**
> > > >
> > > > I still think that the SSL aspect distracts here and is much less relevant than the original task with no additional audio data.

---

> > > > > ### Author Response · Authors · 2022-11-30
> > > > > **Re: Comment**
> > > > >
> > > > > Thank you for your quick response.
> > > > >
> > > > > The recent regularization approach [1] in seq2seq considered not only for the supervised setting but also the SSL-based setting to show their effectiveness.
> > > > >
> > > > > However, we agree that the SSL aspect is much less relevant than the original task. As the reviewer commented above, we will add the new experimental results in the main paper and move the SSL-related works to the appendix.
> > > > >
> > > > >
> > > > > [1] X. Liang et al., “R-drop: regularized dropout for neural networks,” in Proc. NIPS, 2021.

---

> > ### Comment · Reviewer_ngMM · 2022-11-29
> > **Comment**
> >
> > > We have added new experimental results, which can be seen in Appendix D.
> >
> > In my opinion, these should not be in the abstract but must be in the main paper, while all the SSL related work can be moved to the abstract.

---

> > > ### Author Response · Authors · 2022-11-30
> > > **Reply to reviewer ngMM**
> > >
> > > We agree that such modification improves the quality of the paper. Therefore, in the revised manuscript, We will add the new experimental results in the main paper and move the SSL-related works to the appendix.
> > > Thanks for the sincere recommendation.

---

> > ### Comment · Reviewer_ngMM · 2022-11-29
> > **Comment**
> >
> > > We have added new experimental results, which can be seen in Appendix D (pp. 15, 16).
> >
> > The ASR results on librispeech are actually far away from SOTA. The baseline is very weak. Getting some improvements over a weak baseline unfortunately does not tell too much.

---

> > > ### Author Response · Authors · 2022-11-30
> > > **Reply to reviewer ngMM**
> > >
> > > To deal with the baseline performance issue, we additionally have applied beam search decoding with a strong language model (Transformer LM).
> > >
> > > Experiment results on LibriSpeech:
> > >
> > > |Baseline Model|test-clean|test-other|
> > > |:---|:---:|:---:|
> > > |Conformer-CTC Large (teacher for Guided CTC training and SKD)|2.01%|4.69%|
> > > |Conformer-CTC Small (student baseline)|2.85%|8.34%|
> > >
> > > |Method|Additional # of Params. for Teacher|test-clean|test-other|
> > > |:---|:---:|:---:|:---:|
> > > |Baseline|None|2.85%|8.34%|
> > > |Guided CTC training|122 M|2.88%|8.27%|
> > > |SKD|122 M|2.82%|8.00%|
> > > |Ours, EM-Network|1 M|2.70%|7.80%|
> > >
> > > From the results, we confirmed that the proposed method achieved considerable performance improvements with beam-search decoding compared to the other approaches in the fully supervised setting, yielding WER 2.70 % (RERR 5.26 %) on test-clean and WER 7.80 % (RERR 6.47 %) on test-other. The results show that Guided CTC training did not always perform better than the student baseline with the beam-search decoding, meaning that applying LM was more challenging than greedy decoding, as reported in [1]. While the conventional KD frameworks were not sufficiently effective to improve the baseline for such scenario, the EM-Network always improved the performance of the original baseline.
> > > We will add these experimental results in the revised version.
> > > Thank you for careful reading.
> > >
> > > [1] J. W. Yoon, H. Lee, H. Y. Kim, W. I. Cho, and N. S. Kim, “Tutornet: towards flexible knowledge distillation for end-to-end speech recognition,” IEEE/ACM Transactions on Audio, Speech, and Language Processing, 2021.

---

> ### Author Response · Authors · 2022-11-13
> **Third response to reviewer ngMM**
>
> **Concern #6:** Saying that the latent variable in conventional models does not depend on the target y was a bit confusing to me. Because when you actually compute the CTC loss, you actually compute the posterior p(z|x,y) for the loss. … For example: "leveraging the target sequence as the model’s additional training input". This is wrong. In training of CTC or so, surely you leverage the target sequence. You do that when computing p(z|x,y) for the CTC loss. So this is just a new way to compute this posterior. Instead of computing it exactly, it is now approximated by a separate model. But the paper does not really explain this well and is rather explaining it in a somewhat obscure and confusing way.
>
> **Author response:** As shown in Eq. (2), the conventional CTC algorithm computes $P(z|x)$ for the CTC loss, not $P(z|x, y)$. It considers the ground truth as the sole target in training and does not leverage the target sequence as the model’s additional training input. So, it is hard to explain the proposed method with the commented motivation.
>
> However, we acknowledge that our explanations were not clear enough, which might confuse readers. We have revised the descriptions in Section 1(pp. 1, 2) for better understanding our motivation.
>
> [About Motivation] it is difficult to learn the optimal latent variable for the learning task. For example, in the case of ASR, CTC models often converge to sub-optimal alignment distributions and produce over-confident predictions [6, 7, 8]. Since there is an exponential number of possible alignment paths, and the alignment information between source and target sequences is rarely available during training, settling on the optimal alignment is quite challenging. From the feature perspective, powerful representation is important to achieve the desired performance. For the recent NLP studies, including machine translation (MT), a large pre-trained LM is highly required to obtain the contextualized representation [9, 10, 11].
>
> [About CTC-based EM-Network] Modeling the conditional probability of the EM-Network (teacher mode) is determined by whether the latent variable is explicitly defined. The CTC computation adopts the alignment $z$, and it is difficult to settle on the optimal alignment with the conventional framework. The proposed EM-Network computes the posterior $P(z|x,y)$ for the loss, which aims at predicting a better CTC alignment $z$ by leveraging the source and target inputs. Therefore, the CTC model distilled from the EM-Network does not have to consider the exponential number of possible CTC alignments.
>
> [About AED-based EM-Network] For the attention-based encoder-decoder (AED), where there is no explicit latent alignment, it is challenging to directly apply the same training scheme as the EM-Network for CTC. Simply taking the target $y$ as the additional input may cause an obvious but trivial solution, where the model converges with the conditional probability $P(y|x,y) = \delta(y)$. Inspired by the MLM, we present an alternative that employs the masked version of target $\tilde{y}$ as the additional input instead of using the whole target $y$. The EM-Network for AED computes the posterior $P(\tilde{y}|x,y)$ for loss and provides more robust contextualized representations that can benefit the learning task.
>
> **Author action:**
>
> (1) We have revised the descriptions in Section 1(pp. 1, 2) for better understanding our motivation.
>
> (2) We have added the following references:
>
> - H. Liu et al., “Connectionist temporal classification with maximum entropy regularization,” in Proc. NIPS, 2018.
>
> - J. Yu et al., “Fastemit: low-latency streaming asr with sequence-level emission regularization,” in Proc. ICASSP, 2021.
>
> - Y. Miao et al., “Eesen: End-to-end speech recognition using deep rnn models and wfst-based decoding,” in Proc. ASRU, 2015
> ***
> **Works Cited**
>
> [6] H. Liu et al., “Connectionist temporal classification with maximum entropy regularization,” in Proc. NIPS, 2018.
>
> [7] J. Yu et al., “Fastemit: low-latency streaming asr with sequence-level emission regularization,” in Proc. ICASSP, 2021.
>
> [8] Y. Miao et al., “Eesen: End-to-end speech recognition using deep rnn models and wfst-based decoding,” in Proc. ASRU, 2015.
>
> [9] H. Xu, B. Durme, and K. Murray, “Bert, mbert, or bibert? a study on contextualized embeddings for neural machine translation,” in Proc. EMNLP, 2021.
>
> [10] Z. Wu, L. Wu, Q. Meng, Y. Xia, S. Xie, T. Qin, X. Dai, and T. Liu, “Unidrop: a simple yet effective technique to improve transformer without extra cost,” in Proc. NAACL, 2021.
>
> [11] J. Zhu, Y. Xia, L. Wu, D. He, T. Qin, W. Zhou, H. Li, and T. Liu, “Incorporating bert into neural machine translation,” in Proc. ICLR, 2020.

---

> > ### Comment · Reviewer_ngMM · 2022-11-30
> > **Comment**
> >
> > > As shown in Eq. (2), the conventional CTC algorithm computes p(z|x) for the CTC loss, not p(z|x,y).
> >
> > The conventional CTC algorithm calculates both of that: the forward pass calculates p(z|x) and the backward pass computes p(z|x,y). This is the well known forward -backward algorithm. It is used for the gradient.
> >
> > So this is clearly wrong in the paper.
> >
> > And this is a quite crucial central point.

---

> > > ### Author Response · Authors · 2022-11-30
> > > **Reply to reviewer ngMM**
> > >
> > > Thank you very much for your valuable comment. I want to point out that the “forward” pass in our algorithm calculates $P(z|x,y)$ (Eq. (3)) by utilizing the target information $y$ as the model’s input, while the conventional CTC “forward” pass calculates $P(z|x)$ (Eq. (2)). In contrast to the original CTC, the EM-Network leverages the target information as the “model’s additional input”.
> > >
> > > In the conventional CTC framework, as the reviewer commented, the target information is used to search the valid path of the CTC algorithm, and our algorithm follows the same backward scheme of the conventional CTC. However, since the target information is not explicitly used as the model’s input in the original CTC framework, it is hard to formulated the backward pass as $P(z|x,y)$.
> > >
> > > The main difference from the CTC is that the “forward” pass in our algorithm calculates $P(z|x,y)$. Our algorithm utilizes target information not only to calculate loss function in target side (backward pass for CTC) but also to search better CTC alignments (proposed idea). The proposed EM-Network aims at predicting a better CTC alignment $z$ by leveraging the source and target inputs. As mentioned in the paper, the CTC model distilled from the EM-Network does not have to consider the exponential number of possible CTC alignments since the target input is used as guidance to reduce the candidates of $z$. We will add more explanations about this point in the revised version. Thank you again for your time and the valuable feedback.

---

> > > > ### Comment · Reviewer_ngMM · 2022-11-30
> > > > **Comment**
> > > >
> > > > > it is hard to formulated the backward pass as $P(z|x,y)$.
> > > >
> > > > This is wrong. It's not only hard to formulate it this way. It's indeed exactly what is being calculated in the backward pass.
> > > >
> > > > Your statements are all correct for the forward pass.
> > > >
> > > > However, saying that the CTC framework does not explicitly use the targets is misleading. They are explicitly used to define the loss, and they are explicitly calculated for the gradient.
> > > >
> > > > I don't exactly see why it is so relevant that the forward pass does not use the targets in the CTC framework, while in your approach, it does. Why is that relevant? As argued, you anyway compute p(z|x,y) also for CTC.

---

> > > > > ### Author Response · Authors · 2022-11-30
> > > > > **Re: Comment**
> > > > >
> > > > > I am not sure your statement about the backward probability.
> > > > > In the original CTC framework, the forward and backward processes can be formulated as follows:
> > > > >
> > > > > Forward) $\alpha_{t}(s) = \sum_{\mathcal{B(z_{1:t})}=y_{1:s}}p(z_{1:t}|x)=\sum_{\mathcal{B(z_{1:t})}=y_{1:s}} \prod_{t'=1}^{t} p(z_{t'}|x))$
> > > > >
> > > > > Backward) $\beta_{t}(s)=\sum_{\mathcal{B(z_{t:T})}=y_{s:S}}p(z_{t:T}|x)=\sum_{\mathcal{B(z_{t:T})}=y_{s:S}} \prod_{t'=t}^{T} p(z_{t'}|x))$
> > > > >
> > > > > Combined) For any t$\in$[1,T], $p(y|x)=\sum_{\mathcal{B}(z)=y}p(z|x)=\sum_{s=1}^{S} \frac{p(z_{1:t}|x)p(z_{t:T}|x)}{p(z_t|x)}=\sum_{s=1}^{S} \frac{\alpha_{t}(s)\beta_{t}(s)}{p(z_t|x)}$
> > > > >
> > > > > Therefore, both forward and backward passes are required for computing $\sum_{\mathcal{B}(z)=y}p(z|x)$, not $p(z|x,y)$.

---

> > > > > > ### Comment · Reviewer_ngMM · 2022-11-30
> > > > > > **Comment**
> > > > > >
> > > > > > Maybe my explanation was misleading. Anyway, now take the gradient of that, and you will see that this is exactly $p(z|x,y)$ what you get for the gradient, as the targets for the softmax.

---

> > > > > > > ### Author Response · Authors · 2022-11-30
> > > > > > > **Re: Comment**
> > > > > > >
> > > > > > > The term $p(z|x,y)$ is not used in the expression for the gradient of the objective, as shown in the original CTC paper.
> > > > > > >
> > > > > > > Could you please clarify the point about "the gradient of that"?

---

> > > > > > > > ### Comment · Reviewer_ngMM · 2022-11-30
> > > > > > > > **Comment**
> > > > > > > >
> > > > > > > > The gradient of the CTC loss w.r.t. to the logits, or w.r.t. $\log p(z|x)$, this gradient explicitly uses/contains $p(z|x,y)$.
> > > > > > > >
> > > > > > > > Specifically, the gradient for the logits is:
> > > > > > > > $p(z|x,y) - p(z|x)$.
> > > > > > > >
> > > > > > > > The gradient for the log prob is: $-p(z|x,y)$ (maybe I remember the sign wrong)

---

> > > > > > > > > ### Author Response · Authors · 2022-11-30
> > > > > > > > > **Re: Comment**
> > > > > > > > >
> > > > > > > > > As far as we know, most studies (of course including the original paper) related to the CTC did not use the term $p(z|x,y)$ in the expression for the CTC computation. If we use the term $p(z|x,y)$ for explaning the original CTC framework, it may confuse readers.
> > > > > > > > >
> > > > > > > > > As aforementioned, the important thing is that we propose to use the target information by computing the posterior $p(z|x,y)$ in the "forward" pass.
> > > > > > > > > Regarding the "backward" pass, we followed the same scheme as the original CTC. In this process, the target is used to search the valid paths for computation.
> > > > > > > > > By adding the target input in the forward pass of the CTC, we confirmed that the proposed method can learn better CTC alignments compared to the conventional approach.
> > > > > > > > >
> > > > > > > > > As the reveiwer commented, the information about this point needs to be put in the paper more clearly. Based on the comments, we will add more descriptions about the "forward" and "backward" of the original CTC and our algorithm. Thank you for your valuable feedbacks.

---

> > > > > > > > > > ### Comment · Reviewer_ngMM · 2022-11-30
> > > > > > > > > > **Comment**
> > > > > > > > > >
> > > > > > > > > > But still, my open question is, why is it relevant that it is already in the forward pass? Why is it not sufficient to be in the backward pass?
> > > > > > > > > >
> > > > > > > > > > Yes, ultimately, you can always argue that it leads to improvements, but the motivation/derivation is not really so clear. I would revise that whole part.

---

> > > > > > > > > > > ### Author Response · Authors · 2022-11-30
> > > > > > > > > > > **Re: Comment**
> > > > > > > > > > >
> > > > > > > > > > > **Regarding the gradient)** Firstly, the commented gradient for the log prob ($-p(z|x,y)$) is wrong. When the model takes the source input $x$ and outputs $p(z|x)$, the gradient is as follows:
> > > > > > > > > > >
> > > > > > > > > > > $-\frac{\partial \ln p(y|x)}{\partial u_{k}^{t}}=p(z^t=k|x)-\frac{1}{p(z^{t}=k|x)z_{t}} \sum_{{y_{s}}=k} \hat{\alpha_{t}(s)} \hat{\beta_{t}(s)}$
> > > > > > > > > > >
> > > > > > > > > > > where $u_{k}^{t}$ represents the logit for time $t$ and class $k$, $z_{t}=\sum_{s=1}^{S}\frac{\hat{\alpha_{t}(s)} \hat{\beta_{t}(s)}}{p(z^t=y_{s}|x)}$, $\hat{\alpha_{t}(s)}=\frac{\alpha_{t}(s)}{\sum \alpha_{t}(s)}$, and $\hat{\beta_{t}(s)}=\frac{\beta_{t}(s)}{\sum \beta_{t}(s)}$
> > > > > > > > > > >
> > > > > > > > > > > $\frac{\partial \ln p(y|x)}{\partial p(z^{t}=k|x)}=\frac{1}{p(y|x)}\frac{\partial p(y|x)}{\partial p(z^{t}=k|x)}=\frac{1}{\sum_{s}\frac{\hat{\alpha_{t}(s)} \hat{\beta_{t}(s)}}{p(z^{t}=y_{s}|x)}}\frac{\sum_{y_{s}=k}\hat{\alpha_{t}(s)} \hat{\beta_{t}(s)}}{p(z^{t}=k|x)^2}$
> > > > > > > > > > >
> > > > > > > > > > > It is important to note that there is no $p(z|x,y)$ term.
> > > > > > > > > > >
> > > > > > > > > > >
> > > > > > > > > > > Our algorithm takes the source $x$ and target $y$ as the model's inputs and outputs $p(z|x,y)$. In this case, CTC computes $p(y|x,y)=\sum_{\mathcal{B}(z)=y}p(z|x,y)$. Also, the motivation and derivation are sufficient for both forward and backward passes.

---

> > > > > > > > > > > > ### Comment · Reviewer_ngMM · 2022-11-30
> > > > > > > > > > > > **Comment**
> > > > > > > > > > > >
> > > > > > > > > > > > Yes, and we have the equality:
> > > > > > > > > > > >
> > > > > > > > > > > > $p(z^t=k | x,y) = \frac{1}{p(z^{t}=k|x)z_{t}} \sum_{{y_{s}}=k} \hat{\alpha_{t}(s)} \hat{\beta_{t}(s)} = \frac{1}{\sum_{s}\frac{\hat{\alpha_{t}(s)} \hat{\beta_{t}(s)}}{p(z^{t}=y_{s}|x)}}\frac{\sum_{y_{s}=k}\hat{\alpha_{t}(s)} \hat{\beta_{t}(s)}}{p(z^{t}=k|x)^2}$
> > > > > > > > > > > > .
> > > > > > > > > > > > This is exactly what I am saying.

---

> > > > > > > > > > > > > ### Author Response · Authors · 2022-12-01
> > > > > > > > > > > > > **Re: Comment**
> > > > > > > > > > > > >
> > > > > > > > > > > > > **About the first equality)** The first equality $p(z^t=k | x,y) = \frac{1}{p(z^{t}=k|x)z_{t}} \sum_{{y_{s}}=k} \hat{\alpha_{t}(s)} \hat{\beta_{t}(s)}$ does not hold actually. The term $\frac{1}{p(z^{t}=k|x)z_{t}} \sum_{{y_{s}}=k} \hat{\alpha_{t}(s)} \hat{\beta_{t}(s)}$ equals to ${\frac{\partial \ln p(y|x)}{\partial p(z^{t}=k|x)}} p(z^{t}=k|x)$, not $p(z^t=k | x,y)$.
> > > > > > > > > > > > >
> > > > > > > > > > > > > Derivation) ${\frac{\partial \ln p(y|x)}{\partial p(z^{t}=k|x)}} p(z^{t}=k|x) = {\frac{1}{p(y|x)}} {\frac{\partial p(y|x)}{\partial p(z^{t}=k|x)}} p(z^{t}=k|x) = {\frac{1}{p(y|x)}} \sum_{y_{s}=k}\frac{\alpha_{t}(s) \beta_{t}(s)}{p(z^{t}=k|x)^2}p(z^{t}=k|x)$
> > > > > > > > > > > > > $= {\frac{1}{p(y|x)}} \sum_{y_{s}=k}\frac{\alpha_{t}(s) \beta_{t}(s)}{p(z^{t}=k|x)}= \frac{1}{p(z^{t}=k|x)z_{t}} \sum_{{y_{s}}=k} \hat{\alpha_{t}(s)} \hat{\beta_{t}(s)}$
> > > > > > > > > > > > >
> > > > > > > > > > > > > We can confirm that the term $\frac{1}{p(z^{t}=k|x)z_{t}} \sum_{{y_{s}}=k} \hat{\alpha_{t}(s)} \hat{\beta_{t}(s)}$ is calculated from ${\frac{\partial \ln p(y|x)}{\partial p(z^{t}=k|x)}} p(z^{t}=k|x)$, not $p(z^t=k | x,y)$.
> > > > > > > > > > > > >
> > > > > > > > > > > > > We would really appreciate if the reviewer can provide more specific derivation regarding $p(z^t=k | x,y)=\frac{1}{p(z^{t}=k|x)z_{t}} \sum_{{y_{s}}=k} \hat{\alpha_{t}(s)} \hat{\beta_{t}(s)}$.
> > > > > > > > > > > > >
> > > > > > > > > > > > > **About the second equality)** Also, $\frac{1}{p(z^{t}=k|x)z_{t}} \sum_{{y_{s}}=k} \hat{\alpha_{t}(s)} \hat{\beta_{t}(s)}=\frac{1}{\sum_{s}\frac{\hat{\alpha_{t}(s)} \hat{\beta_{t}(s)}}{p(z^{t}=y_{s}|x)}}\frac{\sum_{y_{s}=k}\hat{\alpha_{t}(s)} \hat{\beta_{t}(s)}}{p(z^{t}=k|x)}$, not $\frac{1}{\sum_{s}\frac{\hat{\alpha_{t}(s)} \hat{\beta_{t}(s)}}{p(z^{t}=y_{s}|x)}}\frac{\sum_{y_{s}=k}\hat{\alpha_{t}(s)} \hat{\beta_{t}(s)}}{p(z^{t}=k|x)^2}$.

---

> > > > > > > > > > > > > ### Author Response · Authors · 2022-12-07
> > > > > > > > > > > > > **We are anticipating your feedback**
> > > > > > > > > > > > >
> > > > > > > > > > > > > Dear Reviewer ngMM, since the discussion period will end soon, we would greatly appreciate it if you could take a look at our reply and let us know if you have any remaining questions.
> > > > > > > > > > > > >
> > > > > > > > > > > > > We look forward to addressing any remaining concerns before the end of the discussion period.

---

> > > > > > > > > > > > > > ### Comment · Reviewer_ngMM · 2022-12-08
> > > > > > > > > > > > > > **Comment**
> > > > > > > > > > > > > >
> > > > > > > > > > > > > > I might have made some errors in my equation as I was typing them on phone and from what I only barely remembered. However, my main message still holds: You get $p(z^t | x,y)$ for the CTC gradient.
> > > > > > > > > > > > > >
> > > > > > > > > > > > > > Here is another derivation:
> > > > > > > > > > > > > >
> > > > > > > > > > > > > > $$\frac{\partial \log p(y | x)}{\partial \log p(z^t = k | x)} = \frac{1}{p(y|x)} \cdot \frac{\partial p(y | x)}{\partial \log p(z^t = k | x)}$$
> > > > > > > > > > > > > >
> > > > > > > > > > > > > > We have
> > > > > > > > > > > > > >
> > > > > > > > > > > > > > $$p(y|x) = \sum_{z:y} p(z|x) = \sum_{z:y} \exp \sum_t \log p(z_t | x)$$
> > > > > > > > > > > > > >
> > > > > > > > > > > > > > and thus
> > > > > > > > > > > > > >
> > > > > > > > > > > > > > $$\frac{\partial p(y | x)}{\partial \log p(z^t = k | x)} = \sum_{z:y, z^t = k} p(z | x) = p(y, z^t = k | x)$$
> > > > > > > > > > > > > >
> > > > > > > > > > > > > > Now, via Bayes:
> > > > > > > > > > > > > >
> > > > > > > > > > > > > > $$\frac{1}{p(y|x)} \cdot p(y, z^t = k | x) = p(z^t = k | x,y)$$
> > > > > > > > > > > > > >
> > > > > > > > > > > > > > And that is what I was saying:
> > > > > > > > > > > > > >
> > > > > > > > > > > > > > $$\frac{\partial \log p(y | x)}{\partial \log p(z^t = k | x)} = p(z^t = k | x,y)$$

---

> > > > > > > > > > > > > > > ### Author Response · Authors · 2022-12-08
> > > > > > > > > > > > > > > **Re: Comment**
> > > > > > > > > > > > > > >
> > > > > > > > > > > > > > > We thank Reviewer ngMM for abundant discussions. With the additional commented derivation, we can formulate the original CTC gradient as follows:
> > > > > > > > > > > > > > >
> > > > > > > > > > > > > > > $\frac{\partial L_{CTC}}{\partial u_{k}^{t}}=p(z^t=k|x; \theta)-\underbrace{p(z^t=k|x,y;\theta)}_\text{valid path}$
> > > > > > > > > > > > > > >
> > > > > > > > > > > > > > > where $p(z^t=k|x,y; \theta)$ represents the probability for the valid path $z$, which equals to the target $y$ after removal of all the blank symbols and consecutive repetitions. However, conventional CTC models often converge to sub-optimal alignment distributions [1, 2, 3].
> > > > > > > > > > > > > > >
> > > > > > > > > > > > > > > In the EM-Network framework, the gradient for the proposed distillation between the sequence model and the EM-Network is given by
> > > > > > > > > > > > > > >
> > > > > > > > > > > > > > > $\frac{\partial L_{KD}}{\partial u_{k}^{t}}=p(z^t=k|x; \theta)-\underbrace{p(z^t=k|x,y;\phi)}_\text{EM-Network}$
> > > > > > > > > > > > > > >
> > > > > > > > > > > > > > > On top of the valid path in the original backward, we can additionally consider the knowledge of the EM-Network. As shown in Figure 4 (Section 6),  the alignment of the EM-Network (teacher mode) is more accurate than that of the inner sequence model (student mode) but not always corresponds to the target $y$, making the student can learn the teacher's knowledge more adaptively. From the perspective of knowledge distillation (KD), the probability of the EM-Network acts like the soft label.
> > > > > > > > > > > > > > >
> > > > > > > > > > > > > > > Ultimately, modeling the posterior $p(z|x;\theta)$ is the important factor for the CTC model's performance.
> > > > > > > > > > > > > > > If we can effectively model the posterior $p(z|x,y;\phi)$ in addition to the valid path of the original backward, the CTC model is able to produce more correct alignment compared to the original CTC backward, and thus the parameter $\theta$ can be optimized properly.
> > > > > > > > > > > > > > >
> > > > > > > > > > > > > > > We thank you again for your time and efforts.
> > > > > > > > > > > > > > >
> > > > > > > > > > > > > > > Based on the comments, we will definitely make this clearer (explanations + derivations) in the revised paper.
> > > > > > > > > > > > > > >
> > > > > > > > > > > > > > > [1] H. Liu et al., "Connectionist temporal classification with maximum entropy regularization," in Proc. NIPS, 2018.
> > > > > > > > > > > > > > >
> > > > > > > > > > > > > > > [2] J. Yu et al., "Fastemit: low-latency streaming asr with sequence-level emission regularization," in Proc. ICASSP, 2021.
> > > > > > > > > > > > > > >
> > > > > > > > > > > > > > > [3] Y. Miao et al., "Eesen: End-to-end speech recognition using deep rnn models and wfst-based decoding," in Proc. ASRU, 2015.

---

> > > > > > > > > > > > > > > > ### Comment · Reviewer_ngMM · 2022-12-08
> > > > > > > > > > > > > > > > **Comment**
> > > > > > > > > > > > > > > >
> > > > > > > > > > > > > > > > Yes. I think this perspective is not really addressed in the paper, although it seems very relevant to me. For me, the question is, why is the one kind of posterior $p(z|x,y)$ better than the other. I would like to have this discussed, and maybe further analyzed.

---

> > > > > > > > > > > > > > > > > ### Author Response · Authors · 2022-12-09
> > > > > > > > > > > > > > > > > **Re: Comment**
> > > > > > > > > > > > > > > > >
> > > > > > > > > > > > > > > > > The gradient for the CTC is as follows:
> > > > > > > > > > > > > > > > >
> > > > > > > > > > > > > > > > > $\frac{\partial L_{CTC}}{\partial u_{k}^{t}}=p(z^t=k|x; \theta)-\sigma_{CTC}(k,t)$
> > > > > > > > > > > > > > > > >
> > > > > > > > > > > > > > > > > where $\sigma_{CTC}(k,t) = p(z^t=k|x,y; \theta)$ represents the posterior probability of the $k$-th phoneme at frame $t$, and it is calculated using the forward-backward algorithm. The conventional CTC produces a correct alignment $\sigma_{CTC}(k,t)$  through the forward-backward calculation on the transcription.
> > > > > > > > > > > > > > > > >
> > > > > > > > > > > > > > > > > In the EM-Network framework, the gradient for the proposed distillation between the sequence model and the EM-Network is given by
> > > > > > > > > > > > > > > > >
> > > > > > > > > > > > > > > > > $\frac{\partial L_{KD}}{\partial u_{k}^{t}}=p(z^t=k|x; \theta)-p(z^t=k|x,y;\phi)$
> > > > > > > > > > > > > > > > >
> > > > > > > > > > > > > > > > > where $p(z^t=k|x,y;\phi)$ is the $k$-th phoneme outputs at frame $t$ of the EM-Network model. Different from the original backward that uses the target in the forward-backward calculation (for generating correct alignment), we apply the auxiliary network $\rho$ on top of the original model $\theta$ (where $\rho$+$\theta$=$\phi$), and the target $y$ is fed to the auxiliary network. The auxiliary network performs a fusion of the CTC model's output and the target $y$'s representation and thus can refine the original CTC model's output based on the information of target input.
> > > > > > > > > > > > > > > > > Now, the output of the EM-Network ($\phi$) is more accurate than that of the original CTC model ($\theta$), which can be seen in Figure 4 (Section 6). It means that the output of the EM-Network is the supportive knowledge for training the CTC model.
> > > > > > > > > > > > > > > > >
> > > > > > > > > > > > > > > > >
> > > > > > > > > > > > > > > > > The probability $p(z^{t}=k|x, y;\phi)$ is derived from the parameter $\phi$, and the CTC model is optimized to imitate it, no matter whether it is correct or wrong compared with the transcription. The probability $p(z^t=k|x,y;\phi)$ of the EM-Network acts like the soft label. This is the key difference from the original backward process. From the KD perspective, the performance improvement by using the soft label seems reasonable.
> > > > > > > > > > > > > > > > >
> > > > > > > > > > > > > > > > > In addition to the original CTC backward, the proposed $p(z^t=k|x,y;\phi)$ can provide the additional alignment information in training the CTC alignment.
> > > > > > > > > > > > > > > > >
> > > > > > > > > > > > > > > > > Also, the quality of $\sigma_{CTC}(k,t)$ is the bottleneck of the convergence speed and performance.
> > > > > > > > > > > > > > > > > Thus, properly optimizing the parameter $\theta$ is the important factor for the CTC model's performance.
> > > > > > > > > > > > > > > > > If we can effectively model the posterior $p(z^t=k|x,y;\phi)$, the CTC model is able to produce more accurate alignment compared to the original CTC backward by using the additional information $p(z^{t}=k|x, y;\phi)$, and thus the parameter $\theta$ can be optimized properly.

---

> ### Author Response · Authors · 2022-11-13
> **Fourth response to reviewer ngMM**
>
> **Concern #7** "their shortcoming is that the latent variable captures only the source-side information". I don't really understand this. Surely it must be this way.
>
> **Author action:** We have removed the following sentence: "their shortcoming is that the latent variable captures only the source-side information".
> ***
> **Concern #8:** What is actually meant by "conventional sequence model" or "seq2seq learning"? This should really be clarified further.  … it's about CTC for speech, and AED for translation, so only those two model types, and nothing else. But this should be totally clear from the beginning on in the paper, probably even in the abstract.
>
> **Author action:** We have added more explanations about the model types in the abstract (pp. 1) and introduction (pp. 2).
> ***
> **Concern #9:** What is meant by CE-based objective function? CTC is a CE-based objective function as well. So this statement or categorization does not make sense to me. This is also not really defined anywhere, so this is a critical flaw. From other parts of the paper, I assume that "CE-based objective" is a very strange way of referring to AED based models?
>
> **Author response:** We acknowledge that the term "CE-based objective" is not clear enough, which may confuse readers. Therefore, we have changed the term “CE-based” to “AED”. For example, we have changed “CE-based EM-Network” to “EM-Network for AED”. Thank you for careful reading.
>
> **Author action:**  We have changed the term “CE-based” to “AED”.
> ***
> **Concern #10:** It seems that the CE-based EM-network does not really fit well into the whole mathematical motivation of the paper. Distillation loss, eq 4: Are these the latent variables? The latent variables are discrete, so the equation does not make sense then. Or are these actually the log probs for the latent vars? Or what exactly are those \hat{z}?
>
> **Author response:** [About EM-Network for AED] We have added the theoretical analysis for AED-based EM-Network, which can be seen in. From the analysis, we confirm that the EM-Network for AED can be interpreted from the EM-like perspective and follows the mathematical motivation of the paper.
>
> [About $\hat{z}$ in Eq. 4] $\hat{z}$ is softmax output of the CTC. The argmax value of $\hat{z}$ corresponds to the predicted CTC alignment. Thank you for careful reading.
>
> **Author action:**
>
> (1) We have added the theoretical analysis for AED-based EM-Network, which can be seen in Section 3 (pp. 6).
>
> (2) We have revised the explanation about $\hat{z}$ and added more descriptions about the relationship between $\hat{z}$ and CTC alignment.
> ***
> **Concern #13:** "Latent CTC Alignment. As visualized in Figure 3, we contrasted the latent alignments" - what does this mean? The latent variables are discrete. The figure does not seem to show any discrete values. So what exactly do we see in the figure?
>
> **Author response:** Figure 3 shows the total frame-wise softmax output (frame-wise label probability) of the CTC, and its argmax value corresponds to the predicted CTC alignment. We agree that the term “latent alignment” in Figure 3 was not clear enough, which might confuse readers. We have revised it as the total frame-wise softmax output and added more descriptions about it. Thank you for careful reading.
>
> **Author action:** We have changed the term “latent alignment” to “total frame-wise softmax output”.

---

> ### Author Response · Authors · 2022-11-28
> **Sincerely looking forward to hearing your feedback**
>
> Dear Reviewer ngMM:
>
> Thank you very much for your precious time and valuable comments. We hope our responses have clarified your concerns. Please kindly let us know if you have more questions. We will try our best to answer them before the rebuttal ends.
>
> Best,
>
> Authors

---

> ### Author Response · Authors · 2022-12-06
> **Follow-up Discussion**
>
> Dear Reviewer ngMM,
>
> We wanted to follow up to see if our responses adequately addressed the concerns you raised in your review of our paper. We would be very grateful if you could provide any additional feedback or comments you may have. We are happy to provide further clarifications or explanations if needed.
>
> Thank you very much for your time and consideration!

---

### Official Review · Reviewer_57uE · 2022-10-27

**Confidence:** 2
**Correctness:** 3
**Technical Novelty And Significance:** 2
**Empirical Novelty And Significance:** 2
**Recommendation:** 6

**Clarity, Quality, Novelty And Reproducibility:**

The paper is mostly clear and easy to follow. It seems like an extension of data2vec that is at least novel for speech recognition tasks.
The source is code is provided, it should help with reproducibility.

**Strength And Weaknesses:**

Strengths
- An interesting approach to incorporate the target sequence to inform the latent variable for seq2seq tasks.
- Empirical results show improvement in state-of-the-art on a speech recognition dataset and two machine translation datasets.

Weaknesses
- Since the EM network uses additional parameters over the other seq2seq baselines, it is not clear if the improvements are due to more parameters or learning a better latent variable. It will be helpful to clarify the number of parameters in the EM-network and the seq2seq model, and contrast it with the number of parameters in the baselines.


**Summary Of The Paper:**

The paper introduces a method for seq2seq tasks which incorporates the target sequence itself to learn a latent variable that informs the target prediction. This is done by using an EM-network that conditions on the target, with cross-attention to the regular seq2seq encoder representations, yielding the latent representations. EM-network is trained to predict the source-target alignment using a CTC loss for speech recognition. For machine translation, the masked target is given as input to the EM-network and task is to reconstruct the actual target. The latent representation informed soft predictions are then distilled back into the seq2seq model. While the speech recognition setting seems more novel, the MT setting seems very similar to data2vec.


**Summary Of The Review:**

I find the approach interesting overall, though it feels a small improvement over data2vec. I am recommending a weak accept.

---

> ### Author Response · Authors · 2022-11-13
> **Response to reviewer 57uE**
>
> We are very grateful for thorough and detailed comments of the reviewer, which helped improve the manuscript.
>
> We are uploading (a) our point-by-point response to the comments (below) (response to reviewers) and (b) an updated manuscript with blue-colored texts indicating changes.
> ***
> **Concern #1:** While the speech recognition setting seems more novel, the MT setting seems very similar to data2vec.
>
> **Author response:** For the data2vec, the student consumes the masked version of the input while the teacher uses the original input. In contrast, in the proposed framework for the MT task, the sequence model (student mode) uses the original source as the model’s input, and the EM-Network (teacher mode) consumes the original source input and the masked version of the target input. Different from the data2vec, the student in our method consumes the original source, not the masked one. Also, we apply the masking strategy to the target input, and the target input is only used for the EM-Network.
> ***
> **Concern #2:** Since the EM network uses additional parameters over the other seq2seq baselines, it is not clear if the improvements are due to more parameters or learning a better latent variable. It will be helpful to clarify the number of parameters in the EM-network and the seq2seq model, and contrast it with the number of parameters in the baselines.
>
> **Author response:** We agree that adding such information improves the quality of the paper. We have added experimental settings about the parameters and compared the number of parameters between the best seq2seq baseline and the EM-Network, which can be seen in Appendix C (pp.14, 15). It is important to note that the auxiliary network of the EM-Network is removed during the inference, and its additional computational load is only required for the training procedure. For the inference procedure, the number of the EM-Network’s parameters is the same as that of the seq2seq baseline. Therefore, the performance gain of the EM-Network does not rely on the additional parameters. Thank you for careful reading.
>
> **Author action:** We have added experimental settings (Table 4 in Appendix C) and explanations (pp. 15) about the parameters, which can be seen in Appendix C.
> ***
> **Concern #3:** I find the approach interesting overall, though it feels a small improvement over data2vec.
>
> **Author response:** Firstly, the proposed EM-Network established SOTA performance with the SSL Base setup. Compared to the best baseline data2vec, it yielded RERR 4.3 % for both test-clean and other datasets. Considering that the data2vec already produced promising ASR results with WER 2.78 % (test-clean) and WER 7.02 % (test-other), the EM-Network yielded considerable improvements for the ASR task. Also, the recent ASR study [1] achieved RERR 3.5 % (for test-clean) compared to the previous competing method, indicating that our performance gain with RERR 4.3 % seems quite significant.
> In addition, to further check the effectiveness of the EM-Network, we have performed additional experiments applying the EM-Network to the fully supervised learning setting, which can be seen in Appendix D.2. From the results, we observed that the proposed method achieved considerable WER performance improvement for fully supervised learning. Compared to the previous KD methods using the SOTA ASR model as the teacher, EM-Network achieved better performance on the LibriSpeech test datasets.
>
> **Author action:** We have added new experimental results, which can be seen in Appendix D (pp. 15, 16).
> ***
> **Works Cited**
>
> [1] S. Ding, T. Chen, and Z. Wang, “Audio lottery: speech recognition made ultra-lightweight, noise-robust, and transferable,” in Proc. ICLR, 2022.

---

> ### Author Response · Authors · 2022-11-23
> **Thank you & looking forward to hearing your feedback**
>
> Dear Reviewer 57uE:
>
> Thank you very much for your precious time and valuable comments. We hope our responses have clarified your concerns. Please kindly let us know if you have more questions. We will try our best to answer them before the rebuttal ends.
>
> Best,
>
> Authors

---

> ### Comment · Reviewer_57uE · 2022-12-02
> **Post Author Response**
>
> I thank the authors for their response which resolved my queries. I have also read the other reviews and maintain my (already favorable) rating.

---

> > ### Author Response · Authors · 2022-12-06
> > **Thank you again for your time in reviewing our work.**
> >
> > Dear Reviewer 57uE:
> >
> > We greatly appreciate your time and efforts. We thank you for taking the time to read our response and the updated version of our paper.

---

### Author Response · Authors · 2022-11-16
**Summary of author response - thank you for the insightful feedback**

We are very grateful for thorough and detailed comments of the reviewers, which helped improve the manuscript. We have updated our paper based on the comments of the reviewers. The changes made are summarized below:

**[R1: About the number of parameters]** We have added experimental settings (Table 4 in Appendix C) and explanations (pp. 15) about the parameters, which can be seen in Appendix C.

**[R2: Clarity regarding Figure1 and Figure 2]** We have added pictorial representation to make it easier for the reader to consume the information: Figure 1 and Figure 2. Also, we have added the description about the teacher forcing in the caption of Figure 2.

**[R2: Further study the behavior of the latent variable]** We have added more descriptions on the CTC alignment, which can be seen in Appendix E (pp. 16, 17).

**[R2: About the source code]** We have added more explanations (about the main differences from original Fairseq) in the github profile.

**[R2: Clarity regarding the model types]** We have added more explanations about the model types in the abstract (pp. 1) and introduction (pp. 2).

**[R2: Clarity regarding some terms, such as CE-based, meaning of $\hat{z}$, and latent alignment in Section 6]**

- We have changed the term “CE-based” to “AED”.
- We have revised the explanation about $\hat{z}$ and added more descriptions about the relationship between $\hat{z}$ and CTC alignment.
- We have changed the term “latent alignment” to “total frame-wise softmax output”.

**[R3: Difference between the EM-Network and the deep mutual learning]** We have added more descriptions about the difference from the deep mutual learning in Section1 (pp. 2).

**[R1, R2, R3: (R1) About the performance improvement for ASR / (R2) Experimental results with the fully supervised learning-based models/(R3) Compared with other methods, such as R-Drop and KD]** We have added new experimental results, which can be seen in Appendix D (pp. 15, 16), and it is confirmed that the EM-Network achieved significant improvement compared to the previous methods in all configurations.

**[R2, R3: About the theoretical analysis]**
- We have added explanations about the lower and upper bounds (pp. 6).
- We have added the theoretical analysis for AED-based EM-Network, which can be seen in Section 3 (pp. 6). The detailed derivations can be seen in Appendix B.
- We have added more descriptions about the tight lower bound in Appendix E (pp. 17).

**[R2, R3: Explanations about latent variable and our motivation]** We have revised the descriptions in Section 1(pp. 1, 2) for better understanding about the latent variable and our motivation.

**[R3: Missing references]** We have added the commented references.

(* As abbreviations, we refer to reviewers 57uE as R1, ngMM as R2, and AXAV as R3, respectively.)

We have uploaded the modified version and the modified part is marked in blue.

Best Wish,

Authors

---

> ### Comment · Area_Chair_bk1a · 2022-12-07
> **Problems in derivation of Eq. 10**
>
> The derivation is unnecessarily complicated, as you can get to the fourth line simply from the definition of the CTC loss: $P(y|x;\theta) = \sum_{z \in \mathcal{B}^{-1}} P(z|x; \theta)$
>
> Also, in line four it does not make sense to sum over $y = \mathcal{B}(z)$ as there is only one value of $y$, and the term in the denominator ought to be $P(z|x,y;\theta^{(t)}, \rho^{(t)})$

---

> > ### Author Response · Authors · 2022-12-08
> > **Re: Problems in derivation of Eq. 10**
> >
> > **Author response:** As the area chair commented, we can simply get the fourth line from the definition of the CTC loss. We will remove unnecessary derivations in the revised version.
> >
> > Regarding $y=\mathcal{B}(z)$ and $P(z|x,y;\theta^{t},\rho^{t})$: This is certainly our bad in terms of writing. We will definitely make this clearer in the revised paper.
> >
> > Thanks for your careful reading and sincere recommendation.
> >
> > **Author action:**
> >
> > (1) We will remove unnecessary derivations in the revised version.
> >
> > (2) We will change the term $y=\mathcal{B}(z)$ to $z \in \mathcal{B}^{-1}(y)$.
> >
> > (3) We will fix the typo $P(z|x, y; \phi)$ to $P(z|x,y;\theta^{(t)},\rho^{(t)})$.

---

> ### Author Response · Authors · 2022-12-11
> **Summary of author response (2)**
>
> **[R3: Performance improvements over KD/R-drop methods]** To deal with the performance improvement issue, we have updated new experiment results by varying the tunable parameter $\alpha$ in Eq. (1), which was used to balance the distillation loss. While $\alpha$ was experimentally set to 2 for SSL-based MT models, which can be seen in Appendix F, we found that the fully-supervised MT model achieved the best performance when $\alpha=5$. For competing methods, including Sequence-level KD and R-Drop, we followed configurations of original papers.
>
> Experiments on IWSLT:
>
> |Method|Additional # of Params. for Teacher|En-De|De-En|
> |:---|:---:|:---:|:---:|
> |Baseline|None|27.87|34.18|
> |Sequence-level KD|206 M|29.33|35.97|
> |R-Drop|0 M|29.43|35.99|
> |Ours, EM-Network|1 M|29.94|36.53|
>
>
> **[R2, R3: ASR baseline (for fully-supervised setting)]** We additionally have applied beam search decoding with a strong language model (Transformer LM) to further improve the performance of the ASR baseline.
>
> |Baseline Model|test-clean|test-other|
> |:---|:---:|:---:|
> |Conformer-CTC Large (teacher for Guided CTC training and SKD)|2.01%|4.69%|
> |Conformer-CTC Small (student baseline)|2.85%|8.34%|
>
> |Method|Additional # of Params. for Teacher|test-clean|test-other|
> |:---|:---:|:---:|:---:|
> |Baseline|None|2.85%|8.34%|
> |Guided CTC training|122 M|2.88%|8.27%|
> |SKD|122 M|2.82%|8.00%|
> |Ours, EM-Network|1 M|2.70%|7.80%|
>
> **[R2: Difference from the original CTC backward]** Based on discussions, we will add detailed descriptions about the difference from the original CTC backward.
> The gradient for the CTC is as follows:
>
> $\frac{\partial L_{CTC}}{\partial u_{k}^{t}}=p(z^t=k|x; \theta)-\sigma_{CTC}(k,t)$
>
> where $\sigma_{CTC}(k,t)$ represents the posterior probability of the $k$-th phoneme at frame $t$, and it is calculated using the forward-backward algorithm. The conventional CTC produces a correct alignment $\sigma_{CTC}(k,t)$  through the forward-backward calculation on the transcription.
>
> In the EM-Network framework, the gradient for the proposed distillation between the sequence model and the EM-Network is given by
>
> $\frac{\partial L_{KD}}{\partial u_{k}^{t}}=p(z^t=k|x; \theta)-p(z^t=k|x,y;\phi)$
>
> where $p(z^t=k|x,y;\phi)$ is the $k$-th phoneme outputs at frame $t$ of the EM-Network model. Different from the original backward that uses the target in the forward-backward calculation (for generating correct alignment), we apply the auxiliary network $\rho$ on top of the original model $\theta$ (where $\rho$+$\theta$=$\phi$), and the target $y$ is fed to the auxiliary network. The auxiliary network performs a fusion of the CTC model's output and the target $y$'s representation and thus can refine the original CTC model's output based on the information of target input.
> Now, the output of the EM-Network ($\phi$) is more accurate than that of the original CTC model ($\theta$), which can be seen in Figure 4 (Section 6). It means that the output of the EM-Network is the supportive knowledge for training the CTC model.
>
>
> The probability $p(z^{t}=k|x, y;\phi)$ is derived from the parameter $\phi$, and the CTC model is optimized to imitate it, no matter whether it is correct or wrong compared with the transcription. The probability $p(z^t=k|x,y;\phi)$ of the EM-Network acts like the soft label. This is the key difference from the original backward process. From the KD perspective, the performance improvement by using the soft label seems reasonable.
>
> In addition to the original CTC backward, the proposed $p(z^t=k|x,y;\phi)$ can provide the additional alignment information in training the CTC alignment.
>
> Also, the quality of $\sigma_{CTC}(k,t)$ is the bottleneck of the convergence speed and performance.
> Thus, properly optimizing the parameter $\theta$ is the important factor for the CTC model's performance.
> If we can effectively model the posterior $p(z^t=k|x,y;\phi)$, the CTC model is able to produce more accurate alignment compared to the original CTC backward by using the additional information $p(z^{t}=k|x, y;\phi)$, and thus the parameter $\theta$ can be optimized properly.

---

### Decision · Program_Chairs · 2023-01-20

**Decision:**

Reject

**Justification For Why Not Higher Score:**

This paper enjoyed a lengthy and thorough discussion between the authors and reviewers, and the AC thanks the reviewers for the extra effort they put into this paper.

The primary reason not to score this paper higher is that the reviewers and AC do not find the analogy between the proposed distillation framework and the EM algorithm to be convincing or helpful, and they recommend that if this work is submitted to another venue, the authors remove that discussion and instead discuss the method as a form of distillation, possibly with access to privileged information. A secondary reason for not accepting the paper is that the empirical results on the Librispeech task should be stronger.


**Justification For Why Not Lower Score:**

N/A

**Metareview: Summary, Strengths And Weaknesses:**

# Summary
This paper describes a distillation framework for sequence-to-sequence models that is intended to improve the learned latent representation. The idea is to provide the teacher model with the target label sequence as an _input_ which is accessed via a cross-attention mechanism. Many model parameters are shared between the teacher and student models, and depending on the application, some care may be needed to ensure that the teacher cannot find a trivial solution when given access to the target label sequence. Experiments on speech recognition (Librispeech) and WMT'14 and IWSLT'14 En-De and De-En (machine translation) tasks are presented to show the advantages of the proposed method.

# Strengths
- The basic idea of the paper, trying to improve sequence-to-sequence models via distillation from a model that has access to the ground truth as an input, is clever and well motivated.
- The machine translation results are reasonably strong, though one reviewer would have like bootstrap-based significance tests to verify that they are solid.
- The comparison of training L_org to L_em for the ASR task (Figure 4) makes a convincing case for the distillation idea.

# Weaknesses
- The reviewers and AC did not find the attempt to frame the proposed algorithm in terms of the EM algorithm to be particularly convincing or helpful. Portions of the derivations were confusing, some of the notation was sloppy (though it was improved over the course of the discussion), and there were enough differences in the details for the ASR and MT tasks, such as the need to use MSE loss instead of KL divergence in the CTC model and the need to condition the teacher on masked label sequences instead of the correct sequences, that the overall analogy to EM didn't really hold together.
- The consensus among the reviewers and AC is that the proposed algorithm is better presented as a form of distillation or regularization. The AC suggests that one way to think about this algorithm is as a form of distillation with privileged information (Lopez-Paz et al., "Unifying distillation and privileged information," in Proc. ICLR, 2016, https://arxiv.org/pdf/1511.03643.pdf) in which the privileged information used by the teacher is the target label sequence.
- It would be useful to try to quantify the degree to which this regularization leads to better alignments. For the speech recognition experiments, some of the analyses of alignments performed in Raissi et al., "HMM vs. CTC for automatic speech recognition: Comparison based on full-sum training from scratch", 2022 (https://arxiv.org/pdf/2210.09951.pdf) might be a useful tool.
- The ASR experiments without a self-supervised model using extra data would be more convincing if the results were closer to state-of-the-art.


**Summary Of Ac-Reviewer Meeting:**

N/A